# Interleukin-22 regulates neutrophil recruitment in ulcerative colitis and is associated with resistance to ustekinumab therapy

Polychronis Pavlidis [1,12], Anastasia Tsakmaki[2,12], Eirini Pantazi[1,12], Katherine Li[3], Domenico Cozzetto [4], Jonathan Digby-Bell [1], Feifei Yang[3], Jonathan W. Lo[5], Elena Alberts[1], Ana Caroline Costa Sa[3], Umar Niazi [4], Joshua Friedman [3], Anna K. Long[6], Yuchun Ding[7], Christopher D. Carey[6,7], Christopher Lamb [6,7], Mansoor Saqi[4], Matthew Madgwick[8,9], Leila Gul[5,8,9], Agatha Treveil [8,9], Tamas Korcsmaros [5,8,9], Thomas T. Macdonald[10], Graham M. Lord [1,11], Gavin Bewick [2] & Nick Powell [5] ✉

The function of interleukin-22 (IL-22) in intestinal barrier homeostasis remains controversial. Here, we map the transcriptional landscape regulated by IL-22 in human colonic epithelial organoids and evaluate the biological, functional and clinical significance of the IL-22 mediated pathways in ulcerative colitis (UC). We show that IL-22 regulated pro-inflammatory pathways are involved in microbial recognition, cancer and immune cell chemotaxis; most prominently those involving CXCR2[+] neutrophils. IL-22-mediated transcriptional regulation of CXC-family neutrophil-active chemokine expression is highly conserved across species, is dependent on STAT3 signaling, and is functionally and pathologically important in the recruitment of CXCR2[+] neutrophils into colonic tissue. In UC patients, the magnitude of enrichment of the IL-22 regulated transcripts in colonic biopsies correlates with colonic neutrophil infiltration and is enriched in non-responders to ustekinumab therapy. Our data provide further insights into the biology of IL-22 in human disease and highlight its function in the regulation of pathogenic immune pathways, including neutrophil chemotaxis. The transcriptional networks regulated by IL-22 are functionally and clinically important in UC, impacting patient trajectories and responsiveness to biological intervention.

Ulcerative colitis (UC) is the most common form of inflammatory bowel disease (IBD). It is an incurable, chronic, immune-mediated inflammatory disease (IMID) that selectively affects the colon and is associated with significant complications, including cancer[1,2]. Dysregulated mucosal immune responses are at the heart of UC pathogenesis characterized by increased production of cytokines and the local accumulation of immune cells, most notably mononuclear cells and neutrophils which are associated with architectural distortion of tissue, crypt destruction and crypt abscess formation[3]. Accordingly,

targeting individual cytokines or the cells that produce them are the most effective therapeutic strategies in UC.

Ustekinumab is a monoclonal antibody (mAb) targeting the p40 subunit common to both interleukin (IL)–12 and IL-23[4]. Selective targeting of IL-23 is a conceptually attractive approach, since IL-23 is strongly implicated in IMIDs affecting the skin[5], brain[6], joints[7] and intestine[8,9]. IL-23 overexpressing transgenic mice develop multi-system inflammatory disease, including severe neutrophilic inflammation in the gut[10]. In preclinical models of UC, genetic deletion, or

therapeutic neutralization of the specific p19 subunit of IL-23 significantly attenuates colitis[11,12]. IL-23 stimulates effector function of innate and adaptive lymphocytes, triggering production of IL-17A, IL-17F, interferon-γ (IFNγ) and GM-CSF[13]. Although clinical trials evaluating the efficacy of IL-23 blockade are now underway, important theoretical concerns about targeting IL-23 exist, since the downstream pathways regulated by IL-23 are also implicated in tissue restitution. For instance, IL-22 is one of the key cytokines regulated by IL-23, and several lines of evidence point to IL-22 having an important protective function in the gut. IL-22 induces production of anti-microbial peptides and is involved in intestinal epithelial barrier recovery after acute injury by promoting LGR5[+] intestinal epithelial stem cell proliferation[14]. Clinical trials evaluating recombinant IL-22 to promote recovery of epithelial injury are currently underway (NCT02749630). Confusingly, in several chronic models of IBD, IL-22 has been shown to be pathogenic[15]. Accordingly, new insights into the function of IL22 in mucosal immunity are now needed, especially in human disease, to help reconcile these discrepancies.

In this study, we probe the clinical and functional significance of the IL-22 responsive transcriptional program in diseased tissue of UC patients treated with ustekinumab and in multiple models of colitis. We provide further insights into cytokine-mediated regulation of the intestinal epithelium and how this influences pathogenic pathways and patient outcomes in UC. We show that IL-22 is a functionally important regulator of neutrophil recruitment to the colon by controlling the expression of neutrophil-active CXC-family chemokines. Augmented expression of IL-22 responsive transcripts and increased recruitment of colonic neutrophils is associated with treatment resistance to ustekinumab.

## Results

### Enrichment of IL-22 responsive transcriptional networks is associated with poor response to ustekinumab therapy in ulcerative colitis

As IL-22 selectively targets the intestinal epithelium we generated a human mini-gut colonic epithelial organoid system in order to investigate the IL-22 regulated molecular pathways. Colonic organoids were treated with or without human recombinant IL-22, and the IL-22-responsive transcriptome was mapped by RNA-seq (Fig. 1a). IL-22 induced differential expression of 1251 transcripts (upregulated: 579, downregulated: 672, FDR < 0.01) (Fig. 1b). Significantly upregulated transcripts encoded anti-microbial peptides (REG1A, REG1B), mucins (MUC1, MUC4, MUC12), chemokines (CXCL1, CXCL2, CXCL5, CXCL8), cytokines (TNF, IL-1, IL18, IL33), caspase family members (CASP1, CASP4, CASP5, CASP10), matrix metalloproteinases (MMP1, MMP7, MMP10), enzymes involved in the generation of reactive oxygen species (DUOXA2, NOS2, SOD2), and immunoregulatory molecules, such as SOCS1, SOCS2, SOCS3 and IDO. Pathway analysis identified tumor necrosis factor-alpha (TNF) and IFNγ responses among other pro-inflammatory pathways to be enriched in the IL-22 regulated DEG list (Fig. 1c).

Next, we asked whether core IL-22 regulated transcripts were differentially expressed in diseased tissue of UC patients. Principal component analysis based on the expression of the top 50 most highly upregulated genes induced by IL-22 in colonic biopsies showed separation of patients with active UC from those with quiescent (inactive) disease and healthy controls (previously reposited dataset: GSE59071, Fig. 1d). A similar picture emerged by testing for the activation of this gene set (top 50 upregulated genes by IL-22) using Gene Set Variation Analysis (GSVA) (Fig. 1e). This is an algorithm which tests whether a group (set) of genes is enriched in complex and heterogeneous samples. The enrichment score varies between +1 (upregulated) to −1 (downregulated) and depends on the distribution of gene expression across the samples tested. We validated these findings in endoscopically acquired colonic biopsies sampled from the sigmoid

colon of patients with moderate-to-severe UC ($n = 550$, Supplementary Table 1) enrolled to the UNIFI phase III clinical trial, a randomized, placebo-controlled trial evaluating the efficacy of ustekinumab, a monoclonal antibody (mAb) that blocks the p40 subunit shared by the human IL-12 and IL-23 cytokines. Our analysis demonstrated that the transcriptional program regulated by IL-22 (Mann–Whitney, two-tailed test, $P < 0.0001$, Fig. 1f) was significantly enriched in UC in comparison with non-IBD control subjects.

Although the IL22 responsive transcriptional program was enriched at population level in UC patients, there was considerable variation in magnitude of enrichment. Since it was unlikely that this variation was being driven by differences in disease severity (all patients in the UNIFI trial program had moderate-to-severe UC, with Mayo endoscopy subscores of either 2 or 3), we considered the possibility that this molecular heterogeneity might represent important differences in underlying disease immunobiology. If molecular stratification of UC patients according to the magnitude of IL-22 responsive transcript enrichment was biologically and/or clinically meaningful, we reasoned that UC patients with different degrees of enrichment would experience different outcomes and follow different trajectories. To test this hypothesis, we stratified patients from the UNIFI trial program according to their IL-22 enrichment score (in colonic biopsies sampled immediately prior initiation with ustekinumab) and evaluated whether these differences in molecular phenotype impacted treatment response. Enrichment scores in colonic tissue could differentiate responders and non-responders to ustekinumab induction therapy (including patients on 130 mg and 6 mg/kg dose), (Fig. 1g–i, Supplementary Fig. 1). Remarkably, in comparison with unstratified patients, remission rates in patients with low IL22 enrichment scores (ES < 0) were approximately doubled, including clinical remission (13% vs 25%), mucosal healing (16% vs 26%) and deep remission (a combination of clinical, endoscopic and histologic remission, 12% vs 22%). Conversely, outcomes in UC patients with high IL-22 enrichment scores were broadly comparable to placebo treated patients. In other words, stratification of UC patients according to the magnitude of enrichment of IL-22 responsive transcriptional modules in baseline biopsies sampled at baseline prior to treatment, associates with response to ustekinumab induction therapy.

Interestingly, IL22 enrichment scores were slightly higher in UNIFI participants previously treated with a biologic (anti-TNF), suggesting that high IL-22 enrichment scores may be associated with poor responses to other treatments (Supplementary Fig. 2A). To investigate this possibility, we analyzed four previously published datasets of UC patients treated with other biologics where outcome data and baseline tissue transcriptomic data was available (anti-TNF: GSE23597, GSE16879, GSE92415, vedolizumab: GSE73661). Although there was no significant association of IL-22 enrichment scores with drug response in 3 of these studies, in one study of infliximab treatment, we observed a significant association of IL-22 enrichment scores with primary non-response (Supplementary Fig. 2B). These data suggest that enrichment of IL-22 responsive transcripts in colonic tissue can predict outcomes to ustekinumab and might also portent poor outcomes more generally in UC, including poor responses to other therapies.

### Patient stratification according to the magnitude of enrichment of the IL-22-regulated transcriptional program identifies immunological mechanisms of treatment resistance

Since patients with the greater enrichment of the IL-22-regulated transcriptome were more likely to experience lack of response to ustekinumab, we reasoned that immunological pathways differentiating these patients from those with low IL-22 enrichment might provide insights into mechanisms of treatment resistance. To probe differences in the molecular profile of patients stratified by IL-22 responsive transcripts, we analyzed genome-wide transcript expression changes in biopsies sampled from UC patients from the

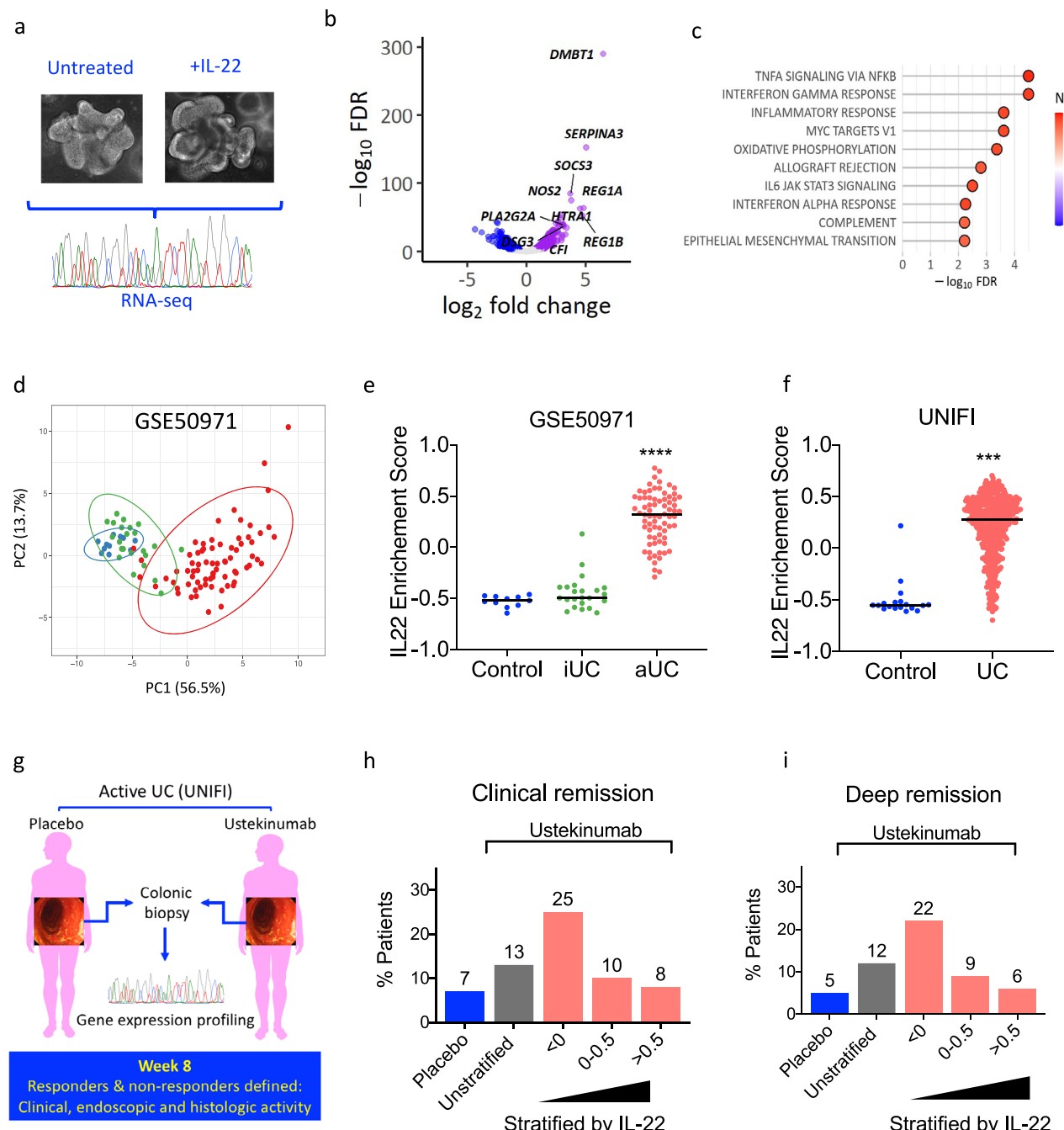

**Fig. 1 | Clinical significance of the IL-22 responsive transcriptional network in ulcerative colitis. a** Experimental schema of IL-22 stimulation of colonic organoids ($n = 4$ biological replicates, IL-22 concentration: 10 ng/ml, duration: 24 h). **b** Volcano plot demonstrating fold change and false discovery rate (FDR) of differentially expressed genes in human colonoids treated with recombinant IL-22. **c** Pathway analysis of DEG regulated by IL-22 (top 10, hallmark gene sets as defined in MSigDB, NES: normalized enrichment score). **d** Expression of the top 50 upregulated transcripts regulated by IL-22 in colonic mucosa separates healthy controls (blue, $n = 11$), patients with endoscopically inactive UC (green, $n = 23$) and active UC (red, $n = 74$) (principal component analysis, reposited dataset: GSE50971). **e** Enrichment of the IL-22 regulated transcriptional program (gene set variation analysis, gene set: top 50 upregulated genes) in the reposited dataset GSE50971 (healthy control: control, iUC-inactive UC, aUC-active UC, Kruskal–Wallis test, ****$p < 0.0001$). **f** Validation of the enrichment for the IL-22 regulated transcriptional program in biopsies taken from healthy controls (control, $n = 18$) and

patients with UC ($n = 550$) participating in UNIFI trial (Mann–Whitney test, two-sided test, ***$p < 0.001$). **g, h, i** Association of the IL-22 enrichment score with clinical outcomes in UNIFI. Clinical remission (defined as a total Mayo score of ≤2 and no subscore >1) and deep remission [which required both clinical remission and mucosal healing defined as histologic improvement (neutrophil infiltration in <5% of crypts, no crypt destruction, and no erosions, ulcerations, or granulation tissue) and endoscopic improvement] at week 8 in UC patients enrolled in the UNIFI clinical trial program stratified according to IL-22 enrichment score in baseline biopsies sampled immediately prior to initiation of ustekinumab ($n = 358$) or placebo ($n = 184$). Blue bar shows response rate in placebo-treated UC patients. Gray bar shows response for all patients treated with ustekinumab. Red bars show response for all patients treated with ustekinumab stratified based on the extent of the IL-22 transcriptional program's activation (ES). Patient numbers and statistical analysis shown in Supplementary Fig. 1. Source data for **b, c, d, e, f, h** and **i** are provided as a Source Data file.

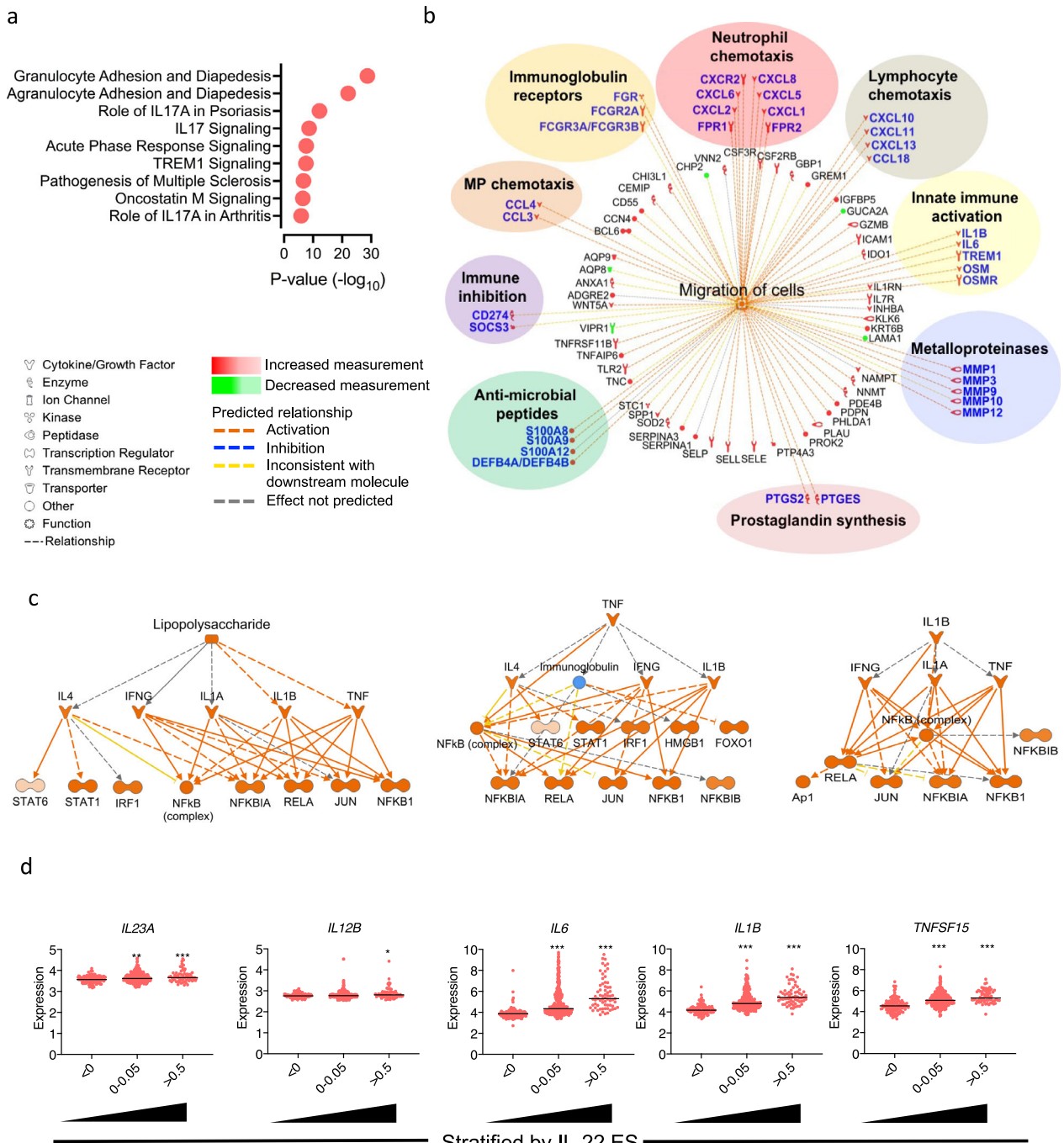

**Fig. 2 | Biological pathways and upstream regulators associated with enrichment of the IL-22 regulated transcriptional program. a** Pathway enrichment analysis (IPA) of the differentially expressed genes in patients enrolled in UNIFI with high enrichment for the IL-22 transcriptional program (right-tailed Fisher's exact test), **b** genes belonging to "Migration of cells" pathway that are upregulated in patients with high enrichment of the IL-22 transcriptional program, as identified using IPA downstream analysis. Genes have been further subdivided into functional categories. Symbols denote type of protein, color denotes regulation and color of line denotes relationship, **c** regulator effects analysis performed on the top 3 predicted upstream activators by IPA in the patients with high enrichment for the IL22 transcriptional program, **d** normalized expression intensity (to healthy controls, $n = 18$) of known IL-22 regulators stratified by the IL-22 transcriptional program enrichment (UNIFI, UC $n = 550$, Kruskal–Wallis test with Dunn multiple comparisons test, *$p < 0.05$, **$p < 0.01$, ***$p < 0.001$). Source data for **a** and **d** are provided as a Source Data file.

UNIFI program with high IL-22 enrichment scores (IL22 ES ≥ 0.25) in comparison with patients with low IL-22 enrichment scores (IL-22 ES < 0.25). Canonical Pathway Analysis (IPA, Ingenuity) demonstrated that the most significantly associated biological pathway in patients with high IL-22 enrichment scores was Granulocyte Adhesion and Diapedesis (right-tailed Fisher's exact test, $P = 1.8 \times 10^{-29}$, Fig. 2a). Other notable associations included multiple Th17 associated pathways, autoimmune diseases, and other pathways that have also previously been linked to treatment resistance including oncostatin M and TREM1 signaling (Fig. 2a and Supplementary Fig. 3). We also performed a Downstream Effects Analysis[16] (IPA, Ingenuity), to predict causal effects and biological processes that were significantly activated in patients with high IL-22 enrichment scores. Overall, there were 245 disease or functional annotations significantly activated in

patients with high IL-22 enrichment scores, encompassing different biological and inflammatory processes. Pathways involving immune cell trafficking were especially activated and comprised the greatest number of functional annotations recorded (Supplementary Fig. 4A, B). In patients with high IL-22 enrichment scores, the highest-ranking causal network associated with cell migration connected 84 nodes, encoding transcripts involved in neutrophil chemotaxis, matrix metalloproteinases, anti-microbial peptides, immunoglobulin Fc receptors and innate-immune response proteins, including IL-1, IL-6, and oncostatin M (Fig. 2b).

To probe which mediators were potentially driving the transcriptional changes observed in patients with ustekinumab resistance, we performed an Upstream Regulator Analysis (IPA, Ingenuity). This algorithm identifies upstream mediators predicted to modulate the expression of transcripts in a user-defined dataset using large-scale causal networks. The top 3 predicted upstream regulators of the gene expression changes observed in colonic biopsies of patients with high IL-22 enrichment scores were lipopolysaccharide (z-score=5.1, right-tailed Fisher's exact test, $P$ value = $2.76 \times 10^{-22}$), TNF (z-score = 4.6, right-tailed Fisher's exact test, $P$ value = $2.97 \times 10^{-18}$), and IL-1β (z-score = 5.7, right-tailed Fisher's exact test, $P$ value = $5.72 \times 10^{-16}$) (Supplementary Fig. 4C). To gauge the biological impact of these predicted mediators we performed Regulator Effects analysis (Ingenuity IPA), an algorithm which connects activated regulators with downstream differentially expressed genes in the dataset. All three of the top predicted upstream regulators had closely related, overlapping mechanistic networks, converging around activation of IL1β, and induction of the transcription factors NFKB1, JUN and RELA (Fig. 2c).

These data also offer insights into unexpected observations in our dataset. We initially anticipated that patients with the highest enrichment scores for IL-22 responsive transcripts would respond favorably to ustekinumab, based on the notion that these patients have augmented IL-23/IL-22 axis activity, and hence are more likely to be amenable to IL-23 blockade, since IL-23 is an important driver of IL-22 production[13,17–21]. However, IL-23 is not the only driver of IL-22 production; other cytokines, such IL-1[22], IL-6[23,24] and TL1A[25] can also trigger IL22 production. Crucially, although there was only a small difference in the expression of transcripts encoding the two subunits of IL-23 (IL-23A and IL-12B) in patients with high IL-22 enrichment scores, there was a substantial increase in the expression of other drivers of IL-22, including *IL1B and IL6* (Fig. 2d, Supplementary Fig. 5). One possible explanation for these observations is that IL-23 blockade with ustekinumab is likely to be ineffective in patients with augmented expression of alternative drivers of IL-22 production, such as IL-1β, and are consistent with the possibility of IL-1β being an important driver of ustekinumab resistance, by triggering activation of IL-22 regulated pathways in an IL-23 independent manner.

### IL-22 regulates expression of neutrophil-active chemokines and other pro-inflammatory transcriptional modules in colonic epithelial cells

Our data imply that IL-22 is potentially involved in mediating a harmful transcriptional program in colonic epithelial cells, and that patients with the greatest magnitude of expression of IL-22 responsive transcripts are likely to be resistant to ustekinumab therapy. To further understand potential pathogenic mechanisms mediated by IL-22, we conducted a more in-depth analysis of IL-22-induced transcriptional changes in colonic organoids at both transcript and pathway level, including a comparison of IL-22 mediated epithelial regulation with changes induced by other cytokines elevated in UC mucosa, including IFNγ, IL-17A, IL-13 and TNF. Canonical Pathway analysis of the IL-22 regulated transcriptional program in colonic epithelial cells confirmed activation of IL-22 signaling and activation of other TREM1 signaling, acute phase response, inflammasome activation, toll-like receptor signaling, Th17 pathway activation and notch signaling (Fig. 3a and Supplementary Fig. 6).

Of the 1251 transcripts regulated by IL-22, 322 (26%) were uniquely regulated by IL-22, whereas 1573 (74%) were additionally regulated by other cytokines. At transcript level, the greatest degree of overlap was observed between IL-22 and IFNγ (Fig. 3b, Supplementary Table 2). There was also substantial overlap between IL-22-regulated transcripts and transcripts regulated by TNF and IL-17A. Conversely, there was relatively low co-regulation of transcripts induced by IL-22 and IL-13. A similar pattern was observed at biological pathway level. The majority of canonical pathways regulated by IL-22 were also regulated by IFNγ and TNF, whereas there was little overlap with IL-13-regulated biological pathways (Fig. 3c).

We also examined whether enrichment scores for these other pro-inflammatory cytokines were associated with response to ustekinumab. In the case of IL-13 and IL-17A, there was no association between their enrichment scores and clinical response, mucosal healing or deep remission (Supplementary Fig. 7). Interestingly, enrichment scores for IFNγ, the cytokine which most strongly overlapped with IL-22, was also associated with response to ustekinumab, albeit slightly less than IL-22. TNF enrichment scores were only weakly associated with some, but not all outcomes in response to ustekinumab.

Biological pathway and causal network analysis identified prominent activation of cell trafficking pathways in IL-22 treated organoids, most notably around neutrophil recruitment (Supplementary Fig. 8). Analysis of the top molecular and cellular functions of the gene expression changes induced by IL22 confirmed a highly significant association with cell movement, which was the most significantly associated annotation. Based on DEGs induced by IL-22, a protein–protein interaction (PPI) network analysis identified cliques of networks regulated by IL-22 in colonoids, and among the most highly complex is a clique of neutrophil attracting chemokines, including CXCL1, CXCL2, CXCL3, CXCL5, CXCL6 and CXCL8 (Fig. 3d) (Analysis in Cytoscape utilizing the STRING database to generate a PPI network of the IL-22 regulated DEG, complexity assessed by the M-Code algorithm. Colors depict fold changes [$\log_e$]).

To further probe how IL-22, and other pro-inflammatory cytokines might impact epithelial regulation of immune cell trafficking we investigated the profile and pattern of chemokine expression in colonoids treated with these different immune cues. Each cytokine studied regulated a unique pattern of chemokine expression (Fig. 3e). IL-22 preferentially upregulated the neutrophil-active CXC-family chemokines *CXCL1, CXCL2, CXCL3, CXCL4, CXCL5, CXCL6* and *CXCL8*, a function shared with IL-17A. IFNγ strongly upregulated *CXCL9* and *CXCL10*, whereas IL-13 was only cytokine to strongly upregulate both eosinophil-active chemokines *CCL24* and *CCL26*. In view of the shared regulation of neutrophil-active chemokines by IL-22 and IL-17A, we further investigated how these 2 cytokines might interact by evaluating gene expression changes occurring in colonic organoids treated with a combination of IL-17A and IL-22. Together IL-17A and IL-22 created synergistic effects for induction of CXC-family neutrophil-active chemokines (Fig. 3f).

### The IL-22 regulated transcriptome correlates with colonic neutrophil accumulation which is associated with resistance to ustekinumab therapy in ulcerative colitis

In keeping with the hypothesis that epithelial-derived IL-22 regulated neutrophil-active chemokines were functionally important in UC pathogenesis, we observed significant upregulation of CXC-family chemokines in the colon of UC patients in comparison with healthy control subjects in 2 independent, large datasets (UNIFI and GSE50971, Fig. 4a, b). Moreover, there was a significant positive correlation observed between the IL-22 enrichment score and the enrichment of CXC-family neutrophil-active chemokines (Fig. 4c).

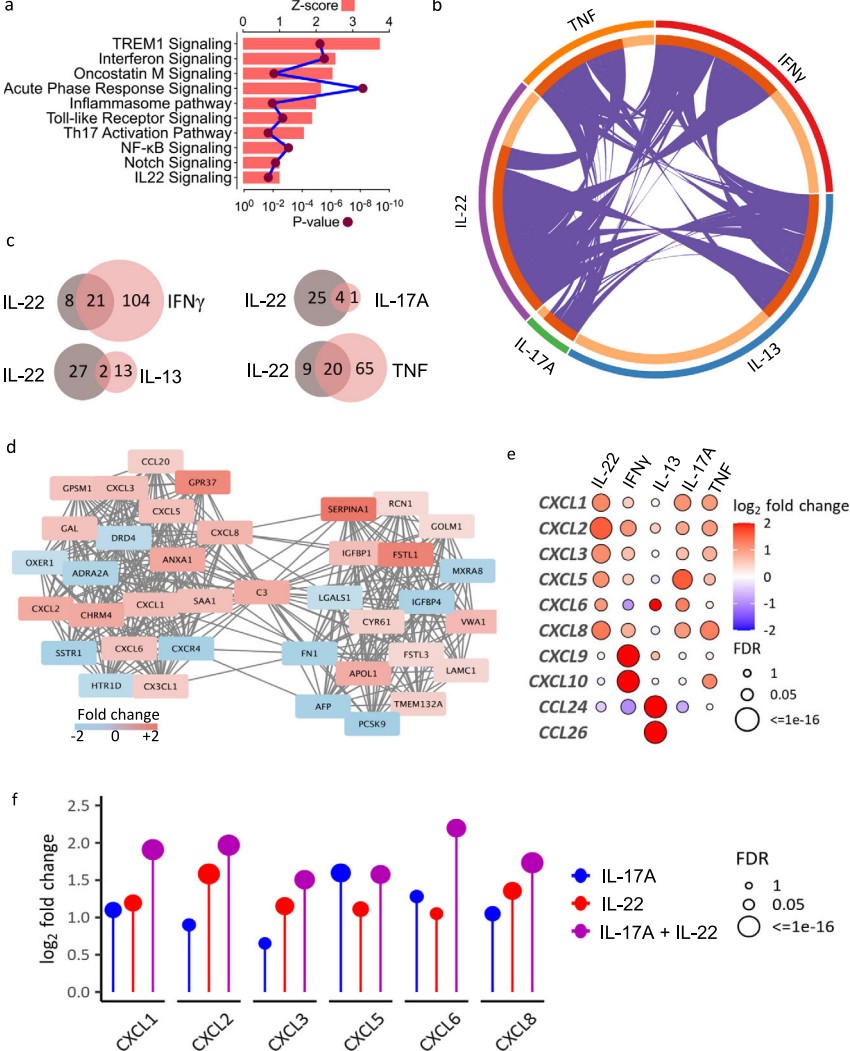

**Fig. 3 | Causal network analysis identifies induction of neutrophil-active chemokines as a key biological activity of IL-22 in the colonic epithelium. a** Top 10 pathways enriched in transcriptional changes regulated by IL-22 in human colonoids (n = 4) as identified by IPA, **b** Circos plot showcasing the shared differentially expressed transcripts regulated by the different cytokines in human colonic organoids (purple lines connecting same genes across DEG lists), **c** Venn diagrams of shared canonical pathways identified in IPA between IL-22 and other pro-inflammatory cytokines, **d** clique of neutrophil attracting chemokines regulated by IL-22 identified by protein–protein interaction (PPI) network analysis (STRING), colors depict fold changes [log$_e$]. **e** Rulation of transcripts coding for chemokines by IL-22 and other cytokines in human colonoids, **f** cumulative effect of IL-22 and IL-17A co-treatment in the expression of neutrophil-attracting chemokines (FDR: false discovery rate). Source data for **a**, **e** and **f** are provided as a Source Data file.

We further investigated this possibility by exploring the relationship between the IL-22 enrichment score and neutrophil recruitment in the colonic mucosa in UC patients in the UNIFI cohort. The histological severity of UC can be scored using haematoxylin and eosin-stained colonic sections using the Geboes score, and severity grading includes an assessment of neutrophil infiltration in both the lamina propria (LP) and epithelium[26]. Strikingly, the magnitude of neutrophil infiltration in the epithelium but not the lamina propria correlated with the magnitude of the IL-22 ES (Fig. 4d).

It is possible to estimate the proportion of different immune cell types in tissue sample based on gene expression profiles and cellular deconvolution tools, such as Cibersort[27]. The estimated proportion of neutrophils in colonic tissue in colonic tissue of UC patients from the UNIFI cohort strongly correlated with IL-22 enrichment scores (Fig. 4e). Moreover, the proportion of neutrophils in colonic tissue also differentiated responders and non-responders to ustekinumab, consistent with the possibility that IL-22-driven colonic neutrophil recruitment has an important function in treatment resistance (Fig. 4f).

A gene set of 57 homolog genes, conserved between mouse and human colitis, has been previously been shown to stratify patients with UC in two groups (UC1, UC2) based on response to biological therapy[28]. In that study, the most highly enriched pathways in patients with refractory disease included neutrophil-related pathways and neutrophil degranulation. We found that 19/57 (33%) of these genes were regulated by IL-22 (21.95 fold enrichment, $P = 1.26e^{-27}$, hypergeometric test), and there was a positive correlation in the enrichment of those 57 genes with the IL-22 ES in the UNIFI cohort (Fig. 4g), providing further supportive evidence of the association between the IL22 transcriptional program, neutrophil chemoattraction and refractory disease.

### IL-22 mediated remodeling of the colonic epithelial transcriptome is conserved across species at gene and pathway level
Next, we investigated whether IL-22-mediated regulation of neutrophil-active chemokine expression was functionally important in colitis. The first step was to evaluate whether IL-22-mediated regulation of human colonic epithelial function was conserved across species. A comparison

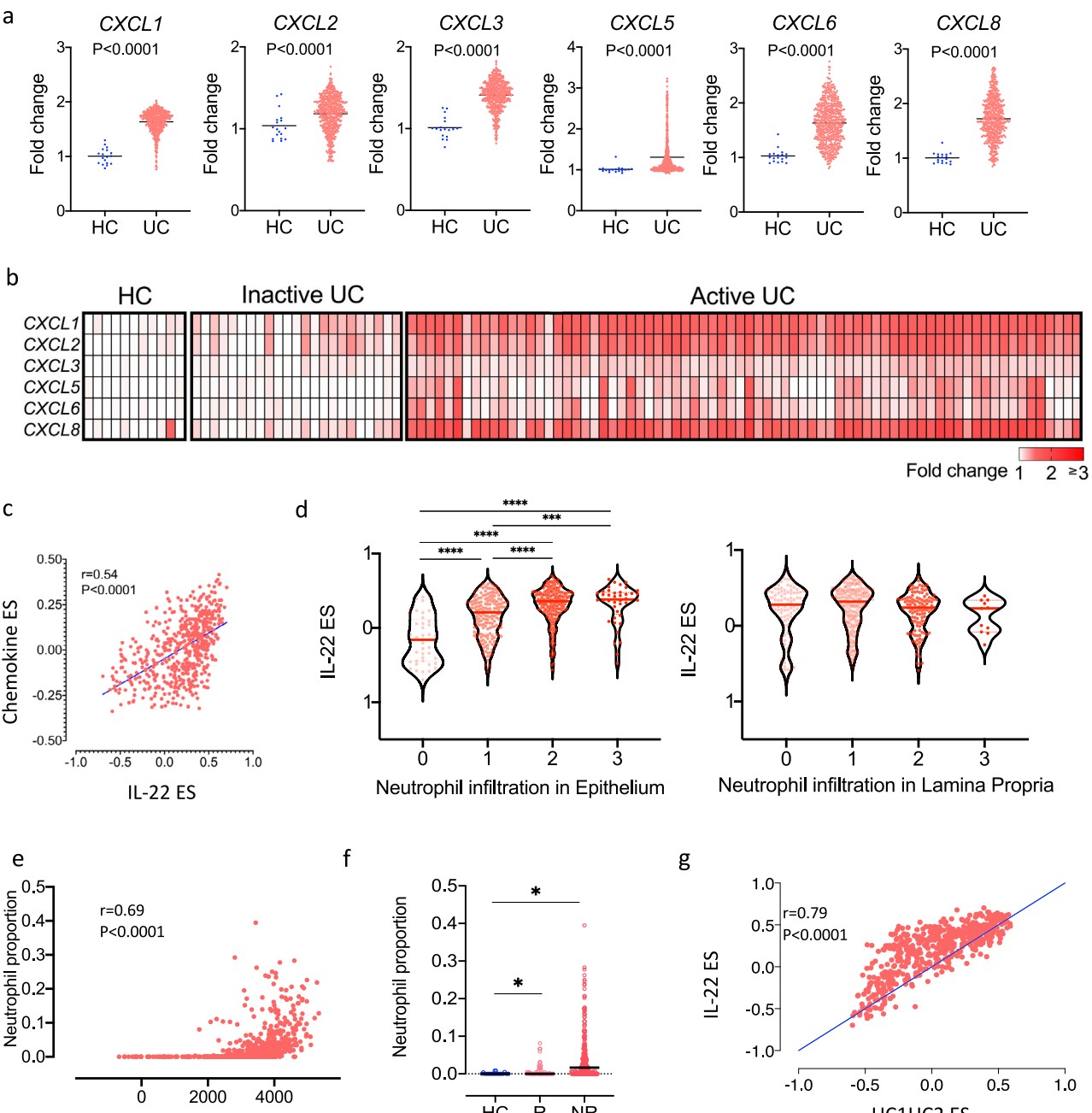

**Fig. 4 | The IL-22 regulated transcriptome correlates with colonic neutrophil accumulation which is associated with resistance to ustekinumab therapy in ulcerative colitis. a** Relative expression of neutrophil attracting chemokines in sigmoid biopsies of UC patients participating to the UNIFI study (*n* = 550) and non-IBD controls (*n* = 18) (Mann–Whitney test, two-tailed, no multiple testing correction applied), **b** relative expression of the neutrophil attracting chemokines in the colonic mucosa of healthy controls (HC), UC patients with inactive and active disease (GSE50971), **c** non-parametric correlation (Spearman two-tailed) between the enrichment score for the IL-22 transcriptional program and the chemokine gene set (*CXCL1, CXCL2, CXCL3, CXCL5, CXCL6, CXCL8*), *r* = 0.54, 95% CI (0.48, 0.60), **d** IL-22 enrichment scores stratified by the neutrophil subscore of the Geboes histology score in the epithelium and lamina propria in colonic biopsies sampled from the UNITI cohort prior to starting ustekinumab. A higher subscore reflects an increase in neutrophil infiltration (for lamina propria 0: no increase, 1: mild but unequivocal increase, 2: moderate increase, 3: marked increase; for epithelium 0: none, 1: <5% crypts involved, 2: <50% crypts involved, 3: >50% involved)

(Kruskal–Wallis test with Dunn's multiple comparisons test, ****p < 0.0001, ***p < 0.001, line denotes median, *n* = 550) **e** bioinformatically computed proportion of neutrophils (Cibersort) in mucosal biopsies of UC patients participating in the UNIFI trial correlates positively (Spearman correlation, two-tailed) with the IL-22 enrichment score (single sample gene set enrichment analysis-ssGSEA), **f** bioinformatically computed proportion of neutrophils (Cibersort) in mucosal biopsies of UC patients participating in the UNIFI trial stratified by response (mucosal healing, week 8) to ustekinumab and healthy controls (HC, *n* = 18). UC: UC patients *n* = 358, R responders, NR non-responders, *p < 0.0001 for NR vs. R and for UC vs. HC, Mann–Whitney test, two-tailed, box & whiskers representing interquartile range and overall range, median value depicted with horizontal line, **g** non-parametric correlation (Spearman, two-tailed) between the enrichment score for the IL-22 transcriptional program and the enrichment score for the gene set defining UC1 and UC2 populations by Czarnewski et al. (*r* = 0.79), 95% CI (0.77,0.83), (FDR false discovery rate). Source data for **a, c, d–g** are provided as a Source Data file.

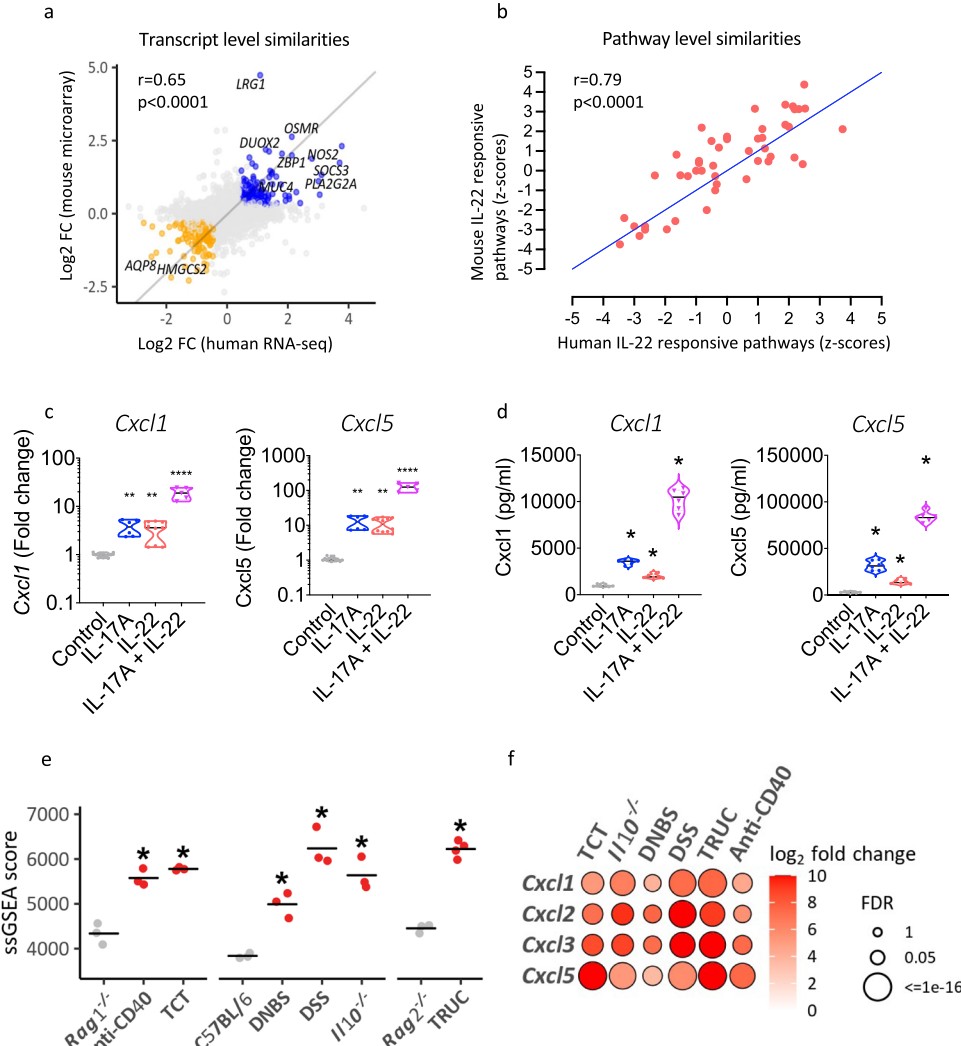

**Fig. 5 | IL-22 mediated remodeling of the colonic epithelial transcriptome is conserved across species at gene and pathway level. a** Non-parametric (Spearman, two-tailed) correlation of transcripts regulated by IL-22 in human and mouse colonoids (FC fold change) [*r* = 0.65, (0.60, 0.71), *p* < 0.0001], **b** non-parametric (Spearman, two tailed) correlation of pathways regulated by IL-22 in human and mouse colonoids (z-scores derived from Ingenuity Pathway Analysis- IPA) [*r* = 0.79, (0.64, 0.88), *p* < 0.0001], **c** effects of IL-22, IL-17A and their combination to the regulation of the neutrophil attracting chemokines *Cxcl1* and *Cxcl5* expression in mouse colonoids, (*n* = 12, Kruskal–Wallis with Dunn's multiple comparisons test, \*\**p* < 0.01, \*\*\*\**p* < 0.0001, median denoted with black line), **d** effects of IL-22, IL-17A and their combination to the production of the neutrophil attracting chemokines Cxcl1 and Cxcl5 in mouse colonoids (*n* = 6, paired Wilcoxon, two tailed test, all comparisons against control, \**p* = 0.31, median denoted with black line), **e** enrichment of the IL-22 transcriptional program (top 50 upregulated genes by IL-22 in mouse colonoids) in preclinical models of colitis (single sample GSEA), including the T-cell transfer model (induced by adoptive transfer of naive CD4[+] T cells to *Rag2*[-/-] mice), the spontaneous, microbiota-dependent models of colitis developing in *Il10*[-/-], and the *Tbx21*[-/-] *Rag2*[-/-] Ulcerative Colitis (TRUC) mice, colitis occurring following administration of DSS administration in drinking water, or rectal administration of DNBS, and innate-immune-mediated colitis occurring in *Rag1*[-/-] mice following administration of agonistic anti-CD40 antibodies (*n* = 3 biological replicates per group, Mann–Whitney test, one-tailed, for all comparisons \**p* = 0.05, besides TRUC vs Rag2[-/-] \**p* = 0.03, line representing median), **f** relative expression of neutrophil attracting chemokines in colonic tissue of different mouse models of colitis. (FDR: false discovery rate). Source data for **a–d** are provided as a Source Data file.

of differentially expressed genes and biological pathways induced by IL-22 in human and mouse colonoids, demonstrated significant correlation at both transcript (Spearman, two-tailed, $r^2 = 0.65$, $P < 0.0001$) and pathway (Spearman, two-tailed, $r^2 = 0.79$ and $P < 0.0001$) level (Fig. 5a, b). In mouse colonic organoids, IL-22 selectively induced expression of the neutrophil-active chemokines *Cxcl1*, *Cxcl3* and *Cxcl5*, with minimal impact on the expression of other CXC-family chemokines (Supplementary Fig. 9A). In the CC family of chemokines, we observed induction of *Ccl7* and weaker induction of *Ccl2* by IL-22, with little or no impact on other chemokine genes (Supplementary Fig. 9B). We validated these findings using real time PCR, which confirmed time and dose-dependent induction of *Cxcl5* (Supplementary Fig. 9C). As observed in human colonoids, IL-22 induced expression of *Cxcl1* and

*Cxcl5*, and was synergistically augmented by IL-17A (Fig. 5c). We confirmed these observations at protein level by measuring chemokine production in supernatants of mouse colonic organoids cultured with recombinant mouse cytokines (Fig. 5d).

We investigated whether IL-22 responsive transcripts were enriched in the colon in mouse models of colitis. GSVA demonstrated significant enrichment of the mouse IL-22 responsive transcriptional module across 6 different colitis models (Fig. 5e). Similar to our observations in human UC, there was significant upregulation of the neutrophil-active chemokines *Cxcl1*, *Cxcl2*, *Cxcl3* and *Cxcl5* across all models of colitis tested, indicating that this core chemokine module is conserved in colitis development across species (Fig. 5f).

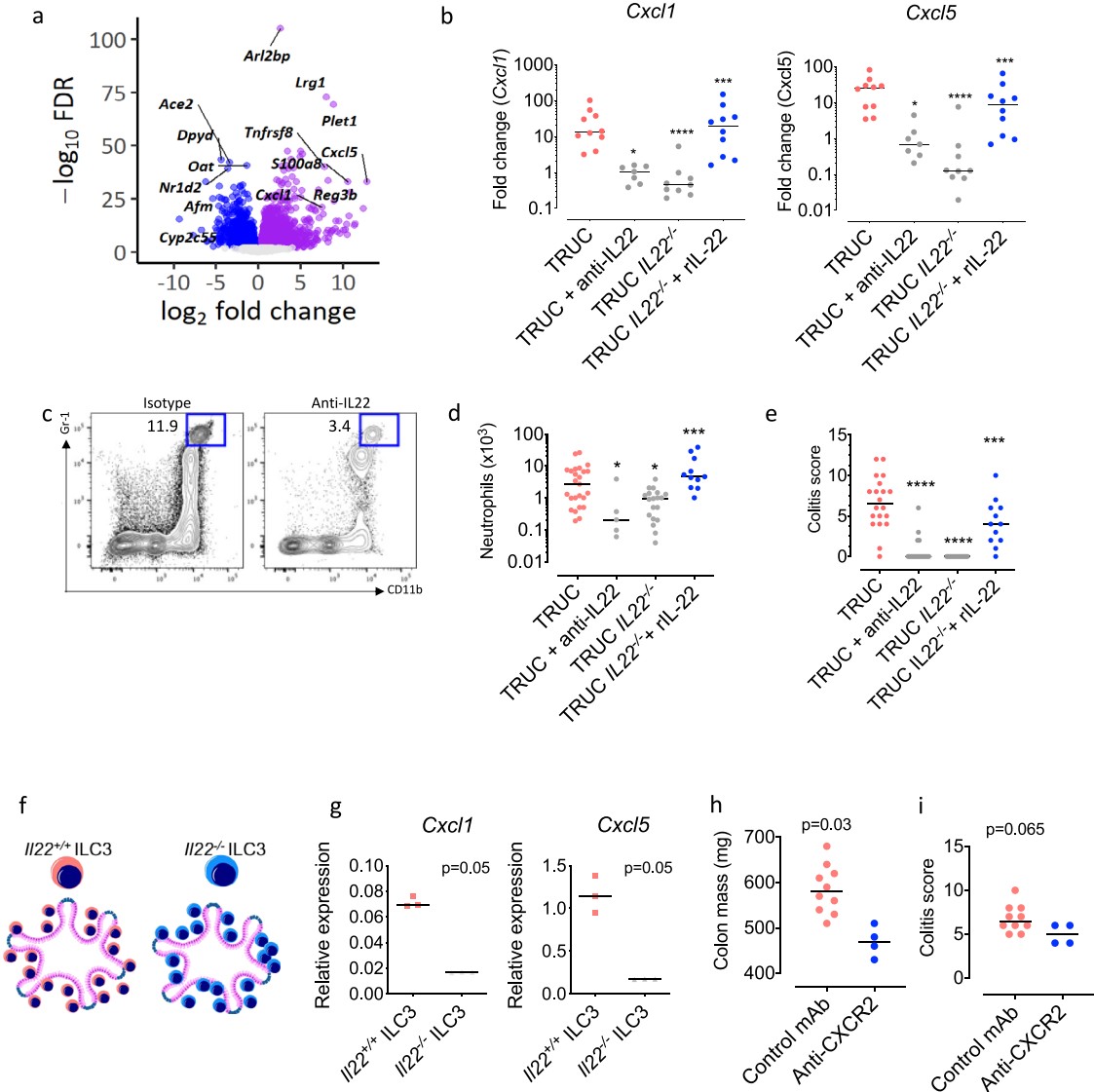

**Fig. 6 | IL-22 is a functionally important regulator of neutrophil recruitment in chronic colitis. a** Volcano plot demonstrating fold change and *P* value of differentially expressed genes in TRUC mice vs controls (Rag2$^{-/-}$)($n$ = 6, biological replicates), **b** effects of IL-22 neutralization to the expression of neutrophil attracting chemokines Cxcl1 and Cxcl5 ($n$ = 36 biological replicates, Kruskal−Wallis with Dunn's multiple comparison test for TRUC + anti-IL22 and TRUC *IL22*$^{-/-}$ *vs* TRUC, *$p$ < 0.05, ****$p$ < 0.0001 and Mann−Whitney two tailed test for TRUC *IL22*$^{-/-}$ + IL-22 *vs*. TRUC *IL22*$^{-/-}$, ***$p$ < 0.001, line denotes median), **c** flow cytometry plot, showing the relative frequency of neutrophils in colonic tissue of TRUC mice treated with a monoclonal antibody blocking IL-22, **d** absolute number of neutrophils in mouse colonic tissue of TRUC mice and TRUC mice with IL-22 neutralization (($n$ = 36, biological replicates, Kruskal−Wallis with Dunn's multiple comparison test for TRUC + anti-IL22 and TRUC *IL22*$^{-/-}$ *vs* TRUC, *$p$ < 0.05, ****$p$ < 0.0001 and Mann−Whitney two tailed test for TRUC *IL22*$^{-/-}$ + IL-22 *vs*. TRUC *IL22*$^{-/-}$, ***$p$ < 0.001,

lines denote median)), **e** effects of IL-22 neutralization on the severity of TRUC colitis ($n$ = 36, biological replicates, Kruskal−Wallis with Dunn's multiple comparison test for TRUC + anti-IL22 and TRUC *IL22*$^{-/-}$ *vs* TRUC, ****$p$ < 0.0001 and Mann−Whitney two tailed test for TRUC *IL22*$^{-/-}$ + IL-22 *vs*. TRUC *IL22*$^{-/-}$, ***$p$ < 0.001, lines denote median), **f** schematic representation of the co-culturing experiment using ILC3 isolated from *Il22*$^{-/-}$ and control mice and cultured with colonoids, **g** relative expression of Cxcl1 and Cxcl5 in colonoids co-cultured with ILC3 derived from *Il22*$^{+/+}$ and *Il22*$^{-/-}$ mice ($n$ = 3, biological replicates per group, Mann−Whitney, one-tailed test, line denotes median). **h** Colon mass and **i** colitis score in TRUC mice treated with anti-CXCR2 ($n$ = 4) or control antibody ($n$ = 14, biological replicates, Mann−Whitney, two-tailed test). Neutralizing anti-IL-22 mAb (clone IL22-01, 200 µg per mouse) were administered intraperitoneally (ip) every 3–4 days. Recombinant IL-22 (rIL-22, 100 µg per mouse) were administered ip at days 0, 4, 8 and 12. Source data for **a**, **b**, **d**, **e**, **g**–**i** are provided as a Source Data file.

## IL-22 is a functionally important regulator of neutrophil recruitment in chronic colitis

Next, we tested the functional significance of IL-22-induced regulation of neutrophil-active chemokines in vivo. *Tbx21*$^{-/-}$ *Rag2*$^{-/-}$ Ulcerative Colitis (TRUC) mice develop chronic, microbiota-dependent colitis with important parallels with human UC. TRUC mice develop chronic, distal colitis which is dependent on IL-23 and TNF[11,29]. Neutrophil-active chemokines were among the most elevated transcripts in the colon of TRUC mice (*Cxcl5* was 2$^{nd}$ and *Cxcl1* the 11$^{th}$ most highly expressed transcripts across the entire genome, Fig. 6a). To test the functional

activity of IL-22 in regulating neutrophil-active chemokines we neutralized, genetically disrupted, or administered recombinant IL22 to TRUC mice. In keeping with IL-22 being an important regulator of neutrophil chemotaxis in the colon, administration of neutralizing anti-IL-22 monoclonal antibody (mAb), or genetic deletion of IL-22 resulted in significant loss of *Cxcl1* and *Cxcl5* expression and a significant reduction in the numbers of neutrophils accumulating in the colon (Fig. 6b–d). Moreover, administration of recombinant (r) IL-22 reinstated *Cxcl1* and *Cxcl5* expression and restored excessive neutrophil recruitment in the colon of TRUC *Il22*$^{-/-}$ mice (Fig. 6b, d).

The functional impact of this axis was also examined by assessing disease activity. IL-22 neutralization or germline deletion of *Il22* was associated with a significant reduction in disease features, including reduced colitis scores and reduced colon mass, whereas administration of rIL-22 restored colitis in otherwise disease free TRUC *Il22*$^{-/-}$ mice (Fig. 6e and Supplementary Fig. 10).

Group 3 innate lymphoid cells are the dominant producers of IL-17A and IL-22 in TRUC disease[11]. Therefore, we purified group 3 ILCs from the colon of TRUC and TRUC *Il22*$^{-/-}$ mice and co-cultured them with mouse colonic organoids (Fig. 6f). Unlike IL-22 sufficient ILC3, which induced *Cxcl1* and *Cxcl5* expression in colonic organoids, induction of these chemokine transcripts was significantly diminished in colonic organoids co-cultured with IL-22 deficient ILC3 (Fig. 6g). This is based on the assumption that organoids were the chief source of chemokines in this experimental system. Although we did not formally test whether ILCs were a potential source of chemokines, previously published work has failed to detect the expression of these chemokines in colonic ILC3s[30].

The functional importance of neutrophil recruitment was further probed by blocking CXCR2, the common receptor expressed by neutrophils for CXC-family neutrophil-active chemokines, including CXCL1 and CXCL5. In vivo administration of anti-CXCR2 mAbs to TRUC mice significantly attenuated TRUC disease (Fig. 6h, i). Taken together, these data support the notion that IL-22-mediated induction of neutrophil-active chemokines, including CXCL1 and CXCL5 is functionally important in the recruitment of CXCR2$^{+}$ neutrophils, and that this pathway has an important pathogenic function in colitis.

### IL-22-mediated induction of neutrophil-active chemokines in colonic epithelial cells is dependent on STAT3 signaling

Next, we sought to define the signaling requirements of IL-22-mediated induction of neutrophil-active chemokine expression. In the intestinal epithelium ligation of IL-22 with its specific receptor triggers activation of different signaling pathways, including STAT3 and MAP kinases, such as MAP3K8[31-34]. Immunostaining confirmed immunoreactivity for IL22RA1 only in the colonic epithelium of healthy controls and patients with UC (Supplementary Fig. 11). Consistent with STAT3 and MAP3K8 having an important function in epithelial signaling in UC, we also observed increased immunostaining for pSTAT3 and MAP3K8 in the epithelial compartment in patients with active UC in comparison to healthy controls (6% ± SEM0.2% *vs.* 19% ± SEM7% and 52% ± SEM19% *vs.* 91% ± SEM6% cells, respectively)(Fig. 7a and Fig Supplementary Fig. 11).

To examine the requirements of STAT3 and MAP3K8 signaling pathways for the IL-22 regulated induction of CXC-family chemokines in colonic organoids, we generated colonoids from mice with epithelial-specific deletion of Stat3 (*Villin*-Cre x *Stat3*$^{fl/fl}$ mice – subsequently termed Stat3$^{\Delta IEL}$), and from mice with germline deletion in MAP3K8. Unlike colonic organoids from control mice (Stat3$^{fl/fl}$), in which IL22 induction of *Cxcl5* was maintained, there was no induction of *Cxcl5* in Stat3$^{\Delta IEL}$ organoids (Fig. 7b). In contrast, IL-22 induction of *Cxcl5* was maintained in *Map3k8*$^{-/-}$ colonoids (Fig. 7c).

To further investigate the dependence of colonic epithelial STAT3 activation for CXC-family chemokine induction in the context of colitis, we analyzed genome-wide changes in the epithelial compartment in DSS colitis, taking advantage of microarray data available from a previously published study[31]. In this study, gene expression profiling was performed on purified colonic epithelial cells from Stat3$^{\Delta IEL}$ and control and mice following induction of colitis. STAT3 responsive genes were defined as transcripts upregulated in epithelial cells from control mice that failed to upregulate in the colonic epithelium of mice with epithelial-specific genetic disruption of STAT3. In Stat3$^{\Delta IEL}$ mice there was no expression of several canonical IL-22-regulated genes, such as *Reg3b*, *Reg3g*, *Fut2* and *Socs3*, consistent with STAT3 being required for the induction of these transcripts in the epithelium.

Moreover, in agreement with our in vitro observations, there was also lower expression of *Cxcl1* and *Cxcl5* colonic epithelium from Stat3$^{\Delta IEL}$ mice (Fig. 7d). These observations have important clinical implications, as small inhibitors of JAK/STAT signaling are now emerging into clinical practice. Tofacitinib, a selective JAK1/JAK3 inhibitor, which prevents phosphorylation and activation of STAT3, has recently been approved for UC[35]. Importantly, tofacitinib significantly inhibited IL-22 induced expression of *CXCL1* and *CXCL5* in human colonoids, indicating that the IL-22/CXC chemokine axis can be therapeutically targeted by JAK/STAT inhibition (Fig. 7e).

## Discussion

This study provides important insights into the immune regulation of the intestinal epithelial barrier with clinically meaningful implications for precision medicine. By mapping IL-22-responsive transcriptional networks in colonic epithelial organoids, evaluating the expression patterns of these genes in colonic tissue, and probing preclinical models of disease, we identify IL-22-mediated regulation of CXCR2$^{+}$ neutrophils as a functionally and prognostically important pathogenic pathway in UC.

The distribution pattern and magnitude of enrichment of IL-22 responsive transcriptional footprints, sampled prior to treatment initiation, could differentiate patients according to their probability of responding to ustekinumab. Although the purpose of this study was not to identify new biomarkers, our data pave the way for exploiting our experimental approach to develop new molecular profiling tools, which could stratify patients as potential responders or non-responders. Precision medicine approaches are much needed in UC to improve patient outcomes, particularly as there are now multiple therapeutic options available[36,37]. Unfortunately, fewer than 50% of patients achieve long-term durable remission with any of the treatment options available, including anti-TNF, anti-integrin, anti-IL12p40 or JAK inhibitors[4,35,38–40]. In this study, UC patients with enrichment scores <0 were more than four times as likely to achieve remission following treatment with ustekinumab in comparison with patients with enrichment scores >0, or placebo-treated patients, and more than twice as likely to achieve remission in comparison with unstratified patients. The notion of harnessing transcriptional signatures to guide treatment strategies is highly attractive[36] and is already exploited in other areas of medicine. To realize the potential of these exciting observations future work will focus on developing robust analytical platforms that may be more appropriate for routine clinical care, including PCR-based technologies.

Analysis of transcriptional modules linked to low and high enrichment of IL-22 responsive transcriptional programs, which were associated with non-response to ustekinumab provided insights into potential mechanisms of ustekinumab resistance. IL-1β was strongly predicted as a potential upstream regulator of the transcriptional changes observed in patients with high IL-22 enrichment scores. Moreover, despite minor differences in the expression of IL-23 subunits, there was a marked increase in the expression of *IL1B*, and to a lesser extent *IL6* and *TL1A* in patients with high IL-22 enrichment scores. Since IL-1β, IL-6 and TL1A are all capable of inducing IL-22 production, it is possible that pathway redundancy and IL-23-independent induction of IL-22 might contribute to ustekinumab resistance in ulcerative colitis.

Causal network analyses of transcripts modulated by IL-22 pointed to a number of potentially important molecular mechanisms of ustekinumab resistance. The most strongly implicated biological pathway in patients with high IL-22 enrichment scores were related to immune cell trafficking and most notably neutrophil recruitment. Others included activation of innate-immune pathways, such as TREM1, which has been linked to resistance to anti-TNF therapy[41]. There was also augmented mucosal expression of metalloproteinases in patients with high IL-22 enrichment scores and ustekinumab

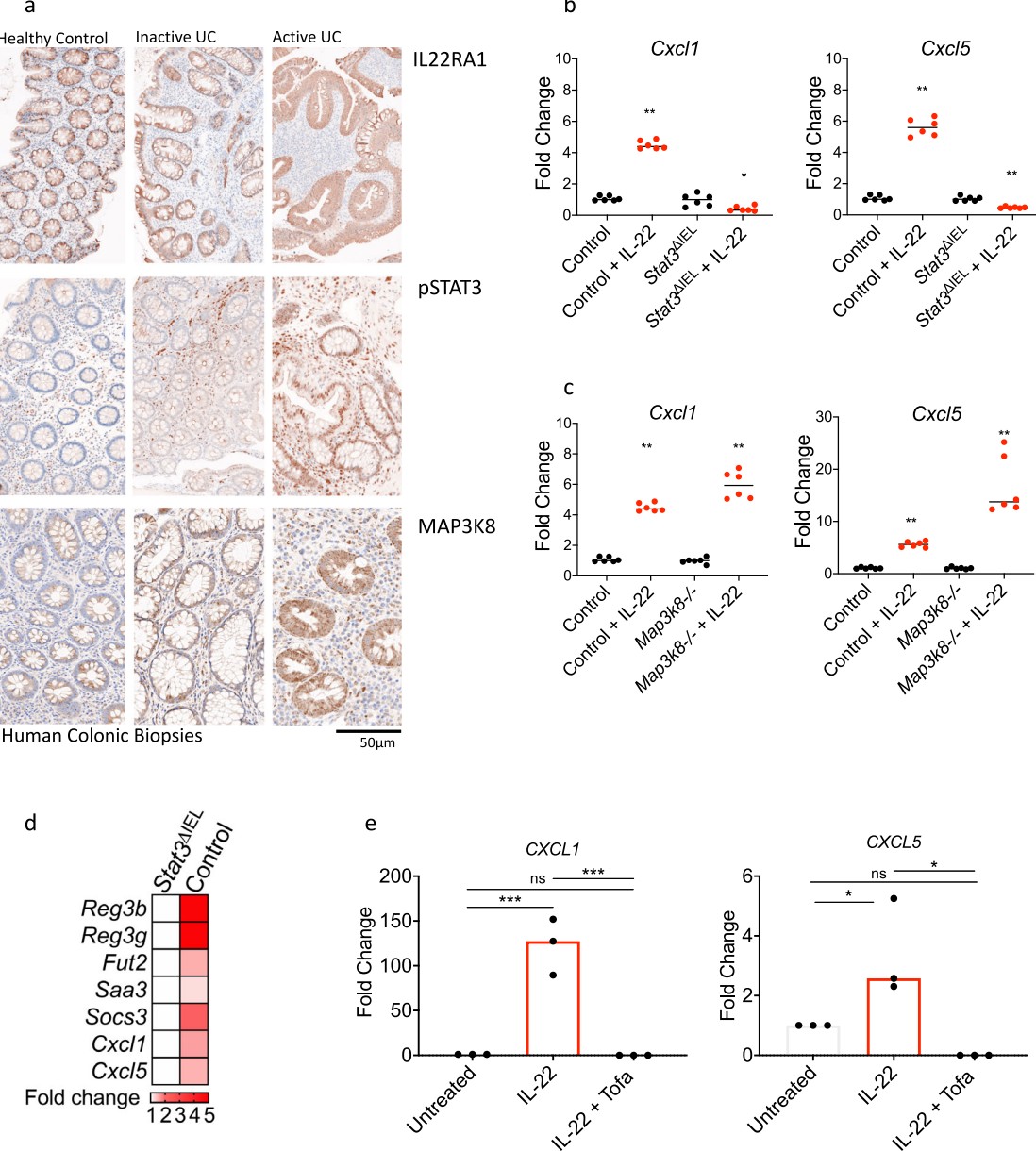

**Fig. 7 | IL-22 mediated induction of neutrophil-active chemokines in colonic epithelial cells is abrogated by JAK inhibition. a** Representative images of immunostaining for IL22RA1, pSTAT3 and MAP3K8 in the colon of a healthy control, UC patient without active inflammation and a UC patients with active inflammation, **b**, **c** relative expression of *Cxcl1 and Cxcl5* in IL-22 treated colonoids generated from *Villin*-Cre x *Stat3*^fl/fl mice (Stat3^ΔIEL), *Map3k8*^−/− mice and controls,

($n = 12$ biological replicates, Mann–Whitney test, two-tailed *$p < 0.05$, **$p < 0.01$). **d** Effects of STAT3 genetic disruption in key gene transcripts regulated by IL-22 (GSE15955) **e** effects of a JAK inhibitor (tofacitinib) in *CXCL1 and CXCL5* expression in human colonic organoids ($n = 3$, biological replicates, ANOVA with Tukey's multiple comparison test, *$p < 0.05$, ***$p < 0.0001$). Source data for **b**–**d** and are provided as a Source Data file.

resistance, a process which has also been linked to resistance to biological therapy, by direct digestion of the therapeutic monoclonal antibody structure through protease activity[42].

Our data address outstanding questions about the function of IL-22 in UC and provide important evidence contrary to current perceptions, indicating that transcriptional programs regulated by IL-22 in UC are likely to be pathogenic. The IL-22-regulated transcriptome was highly conserved between man and mouse and was highly enriched in the colon of both human and mouse colitis. IL-22 mediated regulation of epithelial function was closely shared with other pro-inflammatory cytokines, such as IFNγ and TNF. Immune cell recruitment to sites of inflammation is regulated by local production of chemokines by the inflamed tissue, and by selective expression of chemokine receptors by immune cells. Our data identify the colonic epithelium as a central

communication hub in the regulation of immune cell recruitment to the gut. Different cytokines instructed induction of different chemokine modules by the epithelium, reflecting important qualitative differences in the types of immune responses that are supported by different effector cells. IL-22 was an especially potent inducer of CXC-family neutrophil-active chemokines in colonic organoids, and notably, these chemokines were highly enriched in diseased mucosa of patients with active UC. Importantly, the transcriptional footprint of IL-22 strongly correlated with colonic neutrophil recruitment in the epithelial layer. Moreover, patients failing to respond to ustekinumab therapy had significantly more neutrophils infiltrating the mucosa in comparison with responders, consistent with excessive neutrophil recruitment being an important driver of poor outcomes and treatment resistance. A recent study looking at overlapping transcriptional

features between mice and humans identified a conserved transcriptomic signature that strongly associated with poor outcomes and treatment refractoriness in UC also identified neutrophil-related biological pathways as markers of resistance[28]. There was significant overlap in the transcriptional signature identified by this group, and transcripts that we identified as being IL-22-responsive. As well as corroborating the potential importance of excessive neutrophil accumulation/activation as markers of poor outcomes and treatment resistance in UC, our data provide potential mechanistic explanations for these observations, identifying IL-22 as a potentially important regulator of the "treatment resistance" transcriptional program identified by the Czarnewski study.

Our data also support the notion that IL-22-mediated induction of CXC-family chemokines is functionally important in the recruitment of CXCR2+ neutrophils to the colon in chronic colitis. Blockade, genetic deletion of IL-22, or blockade of CXCR2, the common ligand for the CXC family of neutrophil-active chemokines, significantly attenuated disease in the TRUC model of chronic colitis.

We also defined the signaling requirements for this functionally important colitogenic pathway. Although IL-22 induction of chemokines was maintained in the absence of Map3k8, induction was abolished in the absence of Stat3. With the advent of JAK inhibitors, including agents selectively targeting JAK1, which interacts with STAT3 downstream of the IL22 receptor, it is tempting to speculate that a key therapeutic target of this intervention will be alleviation of IL22-induced CXC-family chemokine induction.

In summary, our study provides further insights into the biology of IL-22 in human disease and highlights how transcriptional networks regulated by IL-22 are functionally and clinically important in UC, impacting patient trajectories and responsiveness to biological intervention. Refinement of this approach could herald a new paradigm for tailoring therapies in UC.

## Methods

The research work described in this paper complies with all ethical regulations relevant to human and animal research. Ethical approval for human samples used for colonoids were provided by King's College London, Guy's and St Thomas' NHS Foundation Trust and King's College Hospital. The national research ethics committee for England reviewed and approved the study protocol (IRAS id:190309). All patients provided samples after informed consent. No compensation was provided. Ethical approval for the immunohistochemistry work on paraffin-embedded tissue of patients was provided by the Newcastle Academic Health Partners Bioresource (Newcastle and North Tyneside 1 REC:12/NE/0395 & 10/H0906/41).

All mice were handled according to local (King's College London) and national guidelines, and all our experimental protocols were reviewed and approved by our local ethics review committee. All animal experiments were conducted in accredited facilities (King's College London, Biological services Unit) in accordance with the UK Animals (Scientific Procedures) Act 1986 (Home Office license number PPL 70/7869).

A list of reagents is provided as supplementary information (Supplementary Table 3).

### Colonoids

Human colonic crypts were isolated from serial colonic biopsies (x2 ascending colon, x2 transverse, x2 descending, x2 rectosigmoid) taken from six adult individuals (median age: 48, range [33,67], female:3), without past medical history or regular medication who attended for routine colonoscopy in view of abdominal symptoms without a diagnosis of IBD and did not have macroscopic or microscopic evidence of inflammation. All patients provided informed consent (NRES/IRAS id: 15/LO/1998). Subsequent establishment of human colonoids was performed as described by Sato et al.[43]. The crypts were cultured

in growth medium containing advanced Dulbecco's modified Eagle's medium/F12, penicillin/streptomycin (100 U/ml), 10 mM HEPES, 2 mM Glutamax, supplements N2 (1×) and B27 (1×), 50 ng/ml mouse epidermal growth factor (all from Life Technologies), 1 mM N-acetylcysteine (Sigma-Aldrich), 50% v/v Wnt3a conditioned medium, 10% v/v R-spondin-1 conditioned medium, 100 ng/ml mouse recombinant noggin protein (Peprotech),10 nM gastrin (Sigma-Aldrich), 500 nM A83-01 (Bio-techne), 10 μM SB202190 (Sigma-Aldrich) and 10 mM Nicotinamide (Sigma-Aldrich). 10 μM Y-27632 (Sigma-Aldrich) was added to the culture medium for the initial 3 days. Medium was changed every 2 days. Differentiation towards a mature epithelium in human colonoids was achieved with reduction of Wnt3a to 15% v/v and withdraw of SB202190 and nicotinamide for 5–7 days. During the last 24 h in differentiation medium human colonoids were treated with human recombinant IL-22 (10 ng/ml), IL-17A (50 ng/ml), TNF (10 ng/ml), IFNγ (20 ng/ml), IL-13 (10 ng/ml) and IL-22 (10 ng/ml)/IL-17A(50 ng/ml) combination. For the experiment presented in Fig. 6 (tofacitinib treated colonoids), tofacitinib was added with IL-22 at the last 24 h of differentiation at a dose of 0.1 μM.

Mouse colonoids were cultured in the same medium as above but without gastrin SB202190, Nicotinamide, A83-01 and with the addition of 3 μM CHIR99021 (Cambridge Biosciences). To differentiate them, Wnt3a was withdrawn for 3 days. During the last 24 h in differentiation medium mouse colonoids were treated with mouse recombinant IL-22 (10 ng/ml) and IL-17A (50 ng/ml).

### Next-generation sequencing and analysis

**RNA extraction.** Colonoids were lysed and RNA was extracted using the RNAeasy kit (Qiagen). This step was optimized balancing the effectiveness of elimination of DNA quantified (Qubit dsDNA HS assay kit) versus the loss of quantity of RNA (Qubit RNA BR assay kit). Optimal DNAse I concentration was determined to be x5 the standard concentration. Five hundred nanograms of cDNA was then created using Revertaid cDNA synthesis kit (ThermoFisher) and diluted to a concentration of 6.25 ng/μl. Harvested colonoids were put in Qiazol and then RNA was extracted with the RNAeasy kit (Qiagen) as per the manufacturer's guidelines. cDNA was created using the Revertaid cDNA synthesis kit (ThermoFisher). Bioanalyzer analysis revealed excellent quality for RNA extracted from both colonoids and whole biopsies (RIN score>9).

**Library preparation and sequencing.** A total amount of 3 μg RNA per sample was used as input material for the RNA sample preparations. Sequencing libraries were generated using NEBNext Ultra RNA Library Prep Kit for Illumina (NEB, USA) following manufacturer's recommendations and index codes were added to attribute sequences to each sample. Briefly, mRNA was purified from total RNA using poly-T oligo-attached magnetic beads. Fragmentation was carried out using divalent cations under elevated temperature in NEBNext First Strand Synthesis Reaction Buffer(5X). First-strand cDNA was synthesized using random hexamer primer and M-MuLV Reverse Transcriptase (RNase H-). Second strand cDNA synthesis was subsequently performed using DNA Polymerase I and RNase H. Remaining overhangs were converted into blunt ends via exonuclease/polymerase activities. After adenylation of 3' ends of DNA fragments, NEBNext Adaptor with hairpin loop structure were ligated to prepare for hybridization. In order to select cDNA fragments of preferentially 150–200 bp in length, the library fragments were purified with AMPure XP system (Beckman Coulter, Beverly, USA). Then 3 μl USER Enzyme (NEB, USA) was used with size-selected, adapter-ligated cDNA at 37 °C for 15 min followed by 5 min at 95 °C before PCR. Then PCR was performed with Phusion High-Fidelity DNA polymerase, Universal PCR primers and Index (X) Primer. At last, PCR products were purified (AMPure XP system) and library quality was assessed on the Agilent Bioanalyzer 2100 system. The clustering of the index-coded samples was performed on a cBot

Cluster Generation System using HiSeq PE Cluster Kit cBot-HS (Illumina) according to the manufacturer's instructions. After cluster generation, the paired-end libraries were sequenced on an Illumina HiSeq platform.

**Gene expression quantification and differential expression analysis.** Fastq files were firstly processed with in-house Perl scripts to discard reads with adapter contamination, or at least 10% of uncertain bases (N), or at least 50% of nucleotides with a Phred quality score less than 20. Read pairs were aligned to the human genome (GRCh37/hg19) using TopHat v2.0.12[44]. HTSeq v0.6.1 was used to count the read pairs mapped uniquely and concordantly to each gene[45]. The raw count matrix was screened for genes with low expression levels across all samples (i.e. average count less than 3), and then with an average number of read pairs less than 3 were filtered out normalized following the strategy suggested by Anders et al.[45].

Differentially expressed genes (DEG) were identified through a varying intercepts hierarchical modeling approach[46–48] implemented in R[48] and Stan[49]. Gene counts were modeled as a negative binomial variable dependent on cytokine treatment as well as covariates accounting for repeated measurements from the same donor and additional sample similarities detected by PCA and hierarchical clustering:

$$\mu_i = \beta_0 + \beta_{t[i]} + \beta_{p[i]} + \beta_{c[i]} \tag{1}$$

$$y_i \sim \text{Negative Binomial}(\exp(\mu_i), \varphi) \tag{2}$$

$$\beta_{[t]} \sim \text{Normal}(0, \sigma_1) \tag{3}$$

$$\beta_{[p]} \sim \text{Normal}(0, \sigma_2) \tag{4}$$

$$\beta_{[c]} \sim \text{Normal}(0, \sigma_3) \tag{5}$$

where:

$t$ = number of groups on treatment (Eq. 3), $p$ = number of subjects (Eq. 4), $c$ = number of clusters (Eq. 5), $i$ = number of observations (Eq. 1), $y$ = gene expression count (Eq. 2), $\varphi$ = overdispersion parameter (Eq. 2), $\sigma_1$ = between treatment standard deviation (Eq. 3), $\sigma_2$ = between-subject standard deviation (Eq. 4), $\sigma_3$ = between cluster standard deviation (Eq. 5).

The quality of the estimated statistical model was assessed through posterior predictive simulations that compare replicated datasets to the actual data. The output p values were corrected for multiple testing with the Benjamini and Hochberg method[50]. Pathway analysis of DEG lists was performed with Ingenuity Pathway Analysis (IPA, Qiagen)[16]. Protein–protein interaction (PPI) analysis was undertaken in Cytoscape[51] utilizing the STRING database[52] to generate a PPI network of the IL-22 regulated DEG and assess complexity by the M-Code algorithm.

**Gene set enrichment analysis.** To test the activation of each of the cytokine-regulated transcriptional programs we used gene set variation analysis (GSVA)[53] and single sample gene set enrichment analysis (ssGSEA) to probe whole transcriptional profiles of previously reposited datasets and the dataset generated in the context of the ustekinumab and golimumab trials programs.

Gene set enrichment analysis (GSEA)[54] was performed using the R Bioconductor package clusterProfiler[55] to test which known pathways are significantly impacted following cytokine treatment. To this end, the genes tested for differential expression between treated and untreated samples were first ranked by decreasing log fold expression changes, and then their enrichment was evaluated against the hallmark

gene sets and KEGG and REACTOME pathway annotations from the MSigDB database[56]. Normalized enrichment scores and empirical $P$ values were estimated using default parameters, and multiple testing correction was carried out using the Benjamini–Hochberg method.

**UNIFI trial program.** The UNIFI trial was a randomized placebo-controlled phase 3 clinical trial evaluating the efficacy and safety of ustekinumab (NCT02407236) and has already been reported[4]. In this study, we report transcriptional data from biopsies, which were correlated to clinical, endoscopic and biomarker data available from the UNIFI cohort (Supplementary Table 1, Demographics). Colonic biopsies were sampled at defined time points (15–20 cm from anal verge) in a subset of patients and were immediately transferred to RNALater (Qiagen) and stored at −80 °C prior to RNA extraction. Whole genome transcriptomics were performed on the Affymetrix HG U133 PM array. Clinical data was recorded prospectively according to the trial protocol. Outcomes reported include: clinical remission (defined as a total Mayo score of ≤2 and no subscore >1) and deep remission [which required both histologic improvement (defined as neutrophil infiltration in <5% of crypts, no crypt destruction, and no erosions, ulcerations, or granulation tissue) and endoscopic improvement] at week 8. The analysis presented is based on all patients receiving ustekinumab regardless of dose (130 mg and 6 mg/kg).

**Reposited datasets.** The following reposited datasets of transcriptomic profiling of patients and healthy controls were accessed and analyzed for this manuscript: GSE59071 ($n = 108$)[57], GSE23597 ($n = 45$)[58], GSE16879 ($n = 24$)[59], GSE92415 ($n = 59$) and GSE73661 ($n = 60$)[60].

**Human colonic biopsy immunohistochemistry.** Immunohistochemistry was performed on formalin-fixed and paraffin-embedded (FFPE) colonic biopsies obtained by colonoscopy from adult patients with histologically active UC ($n = 5$), histologically inactive UC ($n = 5$) and control subjects with no history of UC and no histological inflammation ($n = 5$). Written informed consent was obtained in accordance with research and ethics committee (REC) approval (Newcastle Academic Health Partners Bioresource: Newcastle and North Tyneside 1 REC:12/NE/0395 & 10/H0906/41). 4μm sections of FFPE tissue were stained using a Discovery Ultra autostainer (Ventana Medical Systems, Tucson, AZ) with optimized concentrations of polyclonal anti-IL22RA (1:750 dilution, Novus Biologicals). Slides were developed with 3,3′-Diaminobenzidine (DAB) and counterstained with hematoxylin. Multispectral scanning of stained slides at 10x magnification was undertaken using a Vectra 3.0 Automated Quantitative Pathology Imaging System (PerkinElmer, Hopkinton, MA). InForm Cell Analysis software (v2.0, PerkinElmer) allowed image deconvolution using a spectral library trained on single-stained colonic biopsies for both DAB and Hematoxylin counterstain. Intestinal epithelium and lamina propria were identified by training a tissue segmentation algorithm, which was then applied to all cases, and the accuracy of segmentation was optimized by manual correction. Individual cells in both tissue compartments were identified using a cell segmentation algorithm, based upon the identification of nuclei (hematoxylin). The relative expression of IL22RA1 (membranous/cytoplasmic) was scored in each case and using a binary approach (positive/negative); visual cues were used to distinguish positive staining compared to background, and thresholds were assigned. These data were exported and compiled in MATLAB (v2016b MathWorks, Natick, MA).

### Experimental models of IBD

**Mice.** All mice were housed in specific pathogen-free (SPF) conditions and handled according to local (KCL) and national guidelines. All experimental protocols were reviewed and approved by our local ethics review committee and experiments were conducted in accredited facilities in accordance with the UK Animals (Scientific

Procedures) Act 1986 (Home Office license number PPL 70/7869). Balb/c $Il22^{-/-}$ mice were provided by Pfizer[15]. WT C57Bl/6 and $Rag1^{-/-}$ mice (on C57Bl/6 background) were purchased from Charles River Laboratories. $Il10^{-/-}$ mice were provided by Professor Werner Muller, Faculty of Life Sciences, University of Manchester[61].

## Preclinical models of colitis

**TRUC, TRUC Il22$^{-/-}$.** Balb/c $Tbx21^{-/-}Rag2^{-/-}$ double KO (TRUC) mice (n = 30) develop a communicable, microbiota driven colitis[15,29]. $Tbx21^{-/-}Rag2^{-/-}Il22^{-/-}$ (TRUC $Il22^{-/-}$) triple KO mice (n = 30) were generated by backcrossing Balb/c Tbx21$^{-/-}$Rag2$^{-/-}$ double KO (TRUC) mice with Balb/c Il22$^{-/-}$ mice that were provided by Pfizer.

**aCD40/DNBS.** Two hundred microliters of 3 mg DNBS (Sigma-Aldrich) resolved in 50% EtOH were administered rectally while mice were under isoflurane anesthesia. Mice (n = 3) were monitored daily for weight loss, general signs of distress and adverse disease symptoms. Any mice presented with these features were humanely culled on welfare grounds; otherwise mice were culled 3 days post administration for further analysis.

**DSS.** Three per cent DSS (MW 3600–50,000, MP Biomedicals, LLC) was administered orally in drinking water for 5 or 6 days to C57Bl6 mice and animals were culled at day 7 or 8 respectively. Mice (n = 3) were monitored daily and scored for weight loss, rectal bleeding and feces consistency. Disease activity index was calculated as the sum of the above scores divided by 3[62]. All animals were daily thoroughly observed for general signs of distress or adverse symptoms and any mice presented with these features were humanely culled on welfare grounds.

**Il10$^{-/-}$ mice.** $Il10^{-/-}$ mice (n = 3) were introduced to HT and TRUC microbiota by oral gavage. Mice were monitored twice per week for weight loss, general signs of distress and adverse symptoms. Any mice presented with these features were humanely culled on welfare grounds; otherwise mice were culled 4 weeks post gavage for further analysis.

**TCT.** 0.5 or $2 \times 10^{6}$ naive CD4$^{+}$ T cells (defined as live CD4$^{+}$CD25$^{-}$CD44$^{lo}$CD62L$^{hi}$ cells) were FACS sorted from spleens of 8-week-old C57Bl6 WT female or male donor mice, and injected (in 200 µl of sterile PBS) intraperitoneally into 8–10 week old C57Bl6 $Rag1^{-/-}$ recipients. Purity checks were performed at the end of every sort and cells were always found more than 97% pure. Recipient mice (n = 3) were monitored twice per week for weight loss, general signs of distress and adverse symptoms. Any mice presented with these features were humanely culled on welfare grounds; otherwise, mice were culled 4–6 weeks post adoptive transfer for further analysis.

**In vivo treatments.** Neutralizing anti-IL-22 mAb (clone IL22-01) and recombinant IL-22 (rIL-22) were developed and provided by Pfizer. Two hundred micrograms of IL22-01 (per mouse) were administered ip. every 3 to 4 days. One hundred micrograms of rIL-22 (per mouse) were administered ip. at days 0, 4, 8 and 12, while mice were culled at day 14. Anti-CXCR2 (clone 242216, R&D Systems) was administered ip. at a dose of 100 µg per mouse at days 0, 3, 7, 10 and 14, while mice were culled at day 15.

**Isolation of colonic LP leukocytes (cLPMCs).** Mice were euthanized by either cervical dislocation or by a rising concentration of carbon dioxide gas, and then dissected in a laminar flow cabinet under aseptic conditions. Colons were opened longitudinally, cleaned thoroughly with ice-cold PBS and cut into 1–2 mm pieces and washed with 10 ml 5 mM EDTA, 1 mM Hepes in HBSS (Gibco) in a shaking water bath (300 rpm) at 37 °C for 20 min. Tissue was then vortexed vigorously for

10 sec and passed through a 100 µM cell strainer and collected in C-tubes (Miltenyi) in complete RPMI (Gibco) containing 10% fetal calf serum, 0.25 mg/ml Collagenase D (Roche), 1.5 mg/ml Dispase II (Roche) and 0.01 µg/ml DNase (Roche) and put in a shaking water bath (300 rpm) at 37 °C for 40 min. Before and after the 40 min incubation C-tubes were vigorously shaken for 30 s. Solutions were then passed through 100 µM cell strainers and washed with ice-cold PBS. Cells were resuspended in 10 ml of the 40% fraction of a 40:80 Percoll (GE Healthcare) gradient and carefully placed on top of 5 ml of the 80% fraction in 15 ml tubes. Percoll gradient separation was performed by 20 min centrifugation at $800 \times g$ at room temperature without break. LP cells were collected from the interphase of the gradient and washed with ice-cold PBS. Cells were resuspended in 1 ml PBS, counted and immediately used for further experiments.

**Flow cytometry.** Single cell suspensions were washed with ice-cold PBS and centrifugation at $400 \times g$, 4 °C for 5 min prior to all staining. Cells were then resuspended in 200 µl PBS containing Fc block (aCD32/CD16, eBioscience) at 1:100 dilution and incubated on ice for 10 min. Antibodies against all surface markers were added at appropriate dilutions as well as LIVE/DEAD Fixable Dead Cell Stain (Invitrogen) used in 1:1000. Samples were mixed by vortex and incubated for another 20 min on ice in the dark. After the incubation, cells were washed with ice-cold PBS and centrifugation at $400 \times g$, 4 °C for 5 min and then fixed with 400 µl of 4% PFA and incubated at RT for 15 min in the dark. After fixation, cells were washed again with ice-cold PSB, resuspended in 150–200 µl PBS and stored at 4 °C in the dark awaiting sample acquisition. A representative gating strategy for LP neutrophils is provided in Supplementary Fig. 12. All samples were acquired on a BD LSRFortessa (BD Biosciences) at the Biomedical Research Council (BRC) Flow Core (15th Floor, Tower Wing, Guy's Hospital). Data were analyzed using FlowJo software (Treestar).

**Cell sorting.** To obtain a pure population of naive CD4$^{+}$ T cells from the spleen, splenic single cell suspensions were first treated with ACK buffer for red blood cell lysis, enriched for CD4$^{+}$ cells using immunomagnetic-based cell separation and then stained with mAbs against CD4, CD25, CD44 and CD62L and LIVE/DEAD Fixable Dead Cell Stain (Invitrogen, UK) as described above. Naive CD4$^{+}$ T cells were defined as live CD4$^{+}$CD25$^{-}$CD44$^{lo}$CD62L$^{hi}$ cells. Purity checks were performed after every sort and purity was always found to be above 97%. To obtain pure populations of colonic NCR- ILC3s, cLPMCs were stained with mAbs against CD45, CD90, CD127, KLRG1 and NKp46 and Live/Dead dye as described above. NCR- ILC3s were defined as live CD45$^{+}$CD90$^{+}$CD127$^{+}$KLRG1$^{-}$NKp46$^{-}$ cells (Supplementary Fig. 13). All sorts were performed on BD Aria I, BD Aria II or BD Aria Fusion (BD Biosciences) at the BRC Flow Core (15th Floor, Tower Wing, Guy's Hospital).

**Cell cultures.** Sorted NCR- ILC3s isolated from the colon of TRUC or TRUC Il22$^{-/-}$ mice were cultured for 24 h in complete RPMI (Gibco) containing 10% FCS in the presence or absence of 10 ng/ml IL-23 and 10 ng/ml IL-1β unless stated otherwise. For the co-culture experiments NCR- ILC3s isolated from the colon of TRUC or TRUC Il22$^{-/-}$ mice were activated for 48 h with 20 ng/ml IL-2, 50 ng/ml IL-7, 10 ng/ml IL-23 and 10 ng/ml IL-1β prior to being co-cultured with colonoids.

**ELISA.** Cytokine concentrations in supernatants (S/Ns) of either stimulated cell cultures or explant cultures were measured by ELISA. At the endpoint, S/Ns were harvested and stored at −20 °C pending further analysis. ELISA kits were purchased from eBioscience and ELISAs were performed according to the manufacturer's protocols. Cytokine concentrations were determined within the linear phase of a standard curve made with known cytokine concentrations provided by the supplier.

**Gene expression analysis.** Colonoids or sorted cells were lysed in 1 ml TRIsure (Bioline) and stored at −80 °C pending further processing. Samples were left to thaw at RT and homogenized by vortex for 10 s. To extract the RNA, 200 µl of chloroform were added to each sample followed by 10 s vortex and 15 min incubation at RT. Samples were centrifuged at max speed for 15 min at 4 °C and the clear S/N phase (containing the RNA) was transferred to new 1.5 ml Eppendorf tubes and then mixed with equal volume of isopropanol. Samples were vortex and then left at RT for 10 min, followed by 8 min centrifugation at max speed at 4 °C. RNA pellets were rinsed with 0.5 ml of 75% EtOH and left to airdry at RT. Depending on pellet size; RNA was dissolved in 10-100 µl of RNase/DNase free H2O and stored at −80 °C awaiting further analysis. Concentration of RNA in each sample was measured using NanoDrop. 11 µl of RNA sample (always containing the same amount of RNA across all samples of the same experiment) that was always less that 4 µg RNA, were mixed with 1 µl oligo dT and incubated at 65 °C for 5 min. At the end of the incubation, RNA samples were mixed with 8 µl of reverse transcription mix containing 4 µl Buffer 5×, 1 µl RNase Inhibitor (RI) at 20 U/µl, 2 µl dNTPs and 1 µl Reverse Transcriptase (RT). Reverse transcription was then accomplished by incubating RNA samples at 42 °C for 1 h followed by 65 °C for 5 min and then 4oC forever. cDNA samples (20 µl) were stored at −20 °C until further use. Quantitative PCR was performed using QuantiTect primers (Qiagen) and Quantitect Sybr-Green MasterMix (Qiagen) on a LightCycler 480 (Roche). Samples were analyzed in triplicates and relative expression of mRNAs was determined after normalization against the housekeeping gene Beta-2-Microglobulin (B2M).

**Microarray analysis.** RNA from colonic tissue fragments (distal region) was extracted using TRIsure (Bioline) as described above. Contaminating DNA was removed with the RNase-Free DNase Set (Qiagen) according to the manufacturer's protocol. cDNA was synthetized using Ovation PicoSL WTA System V2 according to the manufacture's protocol (Nugen, USA) and labeled using Encore BiotinIL module according to the manufacture's protocol (Nugen, USA). RNA and cDNA quantity and quality were assessed using the Agilent RNA 6000 Nano Kit or Agilent RNA 6000 Pico Kit (depending on the amount of RNA) according to the manufacture's protocol (Agilent Technologies, USA). Labeled cDNA were hybridized on a MouseWG-6 v2.0 Expression BeadChip (Illumina, USA).

### Statistical analysis

All graphs were generated and analyzed using GraphPad Prism 8 software. Data represent median and interquartile range unless stated otherwise. Statistical analysis was performed using non-parametric Mann–Whitney test or Kruskal–Wallis test unless stated otherwise. Statistical significance was indicated using * for $p$ values less than 0.05, ** for $p$ values less than 0.01 and *** for $p$ values less than 0.001 unless stated otherwise.

### Reporting summary

Further information on research design is available in the Nature Research Reporting Summary linked to this article.

## Data availability

The human organoid data generated in this study have been deposited in the Gene Expression Omnibus database under accession codes GSE190705 and GSE190634. The mouse colitis models data generated in this study have been deposited in the Gene Expression Omnibus database under accession code GSE208395. The UNIFI microarray data have been deposited in the Gene Expression Omnibus database under accession code GSE206285. The following reposited datasets of transcriptomic profiling of patients and healthy controls were accessed and analyzed for this manuscript: GSE59071 ($n = 108$)[57], GSE23597 ($n = 45$)[58], GSE16879 ($n = 24$)[59], GSE92415 ($n = 59$) and GSE73661 ($n = 60$)[60]. Source data are provided with this paper.

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

## Acknowledgements

This work was supported by the Wellcome Trust (WT101159, N.P.), Crohn's and Colitis UK (N.P., P.P.), GutsUK (N.P., P.P.) and KHP Challenge Fund (N.P., P.P.). NP is supported by the NIHR Imperial Biomedical Research Centre (BRC). Additional support is acknowledged from Biotechnological and Biosciences Research Council (BBSRC) Norwich Research Park Biosciences Doctoral Training Partnership (grant numbers: BB/M011216/1 and BB/S50743X/1, recipients: L.G., T.K.). T.K. was supported by the Earlham Institute (Norwick, UK) in partnership with the

Quadram Institute (Norwich, UK) and strategically supported by the UKRI BBSRC UK. This work was also supported by the MRC/ESPRC Newcastle Molecular Pathology Node. We acknowledge the contributions of the National Institute for Health Research (NIHR) Newcastle Biomedical Research Centre (BRC). We would like to thank our collaborators Mark Wilson, NIH (provision of *Map3k8*$^{-/-}$ colonic tissue) and Christoph Becker (villin-cre x *Stat3*$^{fl/fl}$ colonic tissue) to generate relevant knockout organoids. We would like to thank the NIHR BRC Translational Bioinformatics Team for providing access to the Ingenuity Pathway Analysis platform.

## Author contributions

All authors have made substantial contributions to the conception or design of the work; or the acquisition, analysis, or interpretation of data; or have drafted the work or substantively revised it. They have approved the submitted version and to be accountable for this work. PP recruited patients, gathered data, designed and performed experiments, analyzed and interpreted data and contributed to manuscript drafting and revision, AT, EP designed and performed experiments and analyzed data. K.L., J.D.B., D.C., F.Y., J.L., E.A., U.N., J.F., A.C.C.S., A.K.L., Y.D., C.C., C.L., M.S., M.M., L.G., A.T., T.K., T.T.M., G.M.L., G.B. provided intellectual input, performed experiments or analyzed data. N.P. conceptualized the study, supervised the project, analyzed/interpreted data and wrote the manuscript.

## Competing interests

K.L., F.Y., A.C.C.S., J.F. were all employed by Janssen Pharmaceuticals. All other authors declare no competing interests.

## Additional information

[1]School of Immunology and Microbial Sciences, King's College London, London, UK. [2]Diabetes Research Group, School of Life Course Sciences, Faculty of Life Science and Medicine, King's College London, London, UK. [3]Janssen Research & Development, 1400 McKean Rd, Spring House, PA 19477, USA. [4]Translational Bioinformatics, National Institute for Health Research Biomedical Centre, Guy's and St Thomas' NHS Foundation Trust and King's College London, London, UK. [5]Division of Digestive Diseases, Faculty of Medicine, Imperial College London, London, UK. [6]Newcastle upon Tyne Hospitals NHS Foundation Trust, Newcastle upon Tyne, UK. [7]Translational and Clinical Research Institute, Newcastle University, Framlington Place, Newcastle upon Tyne, UK . [8]Earlham Institute, Norwich Research Park, Norwich, UK. [9]Quadram Institute Bioscience, Norwich Research Park, Norwich, UK. [10]Centre for Immunobiology, Barts and the London School of Medicine and Dentistry, QMUL, London, UK. [11]Faculty of Biology, Medicine and Health, University of Manchester, Manchester, UK. [12]These authors contributed equally: Polychronis Pavlidis, Anastasia Tsakmaki, Eirini Pantazi. ✉e-mail: npowell@ic.ac.uk

