## [Peer Review File · Nature Communications]

Interleukin 22 regulates neutrophil recruitment in ulcerative colitis and is associated with resistance to ustekinumab therapyREVIEWER COMMENTS

Reviewer #1 (Remarks to the Author):

In this article, Pavlidis et al established a novel and well thought out method for stratifying patients based on the transcriptomic signature triggered by IL-23 and IL-22. First, the authors obtained these gene signatures using either IL23-stimulated LP mononuclear cells or IL22-stimulated human colonic organoids. Using Gene Set Variation Analysis (GSVA, a sample-specific version of the well-known Gene Set Enrichment Analysis (GSEA)), they evaluated the enrichment of these signature genes in ulcerative colitis patients and healthy controls and correlate the enrichment score (ES) with response or non-response to ustekinumab. Counterintuitively, perhaps in their most important finding, they found that a high IL-22 ES correlated with poor response to ustekinumab. In order to gain mechanistical insights, the authors reproduced the high IL-22 ES in 6 separate mouse models of UC. The authors also investigated possible upstream and downstream mediators of this response and presented several pieces of evidence implicating the role of neutrophil recruitment in the detrimental effects of IL-22. The study holds great potential for improving treatment in ulcerative colitis, both through understanding of pathological events and opening up avenues for personalized treatment through stratification of patients based on colonic biopsies. However, the story overall is disjointed: The results indicating the IL22 downstream effects (eg the role of chemokines, role of STAT3) is somehow disconnected to the most interesting finding which is the identification of IL-1 β as a potential cause of IL-22 ES and lack of responsiveness to ustekinumab. In particular, the relevance of IL17A is not clear and rather muddles the message of the paper. The paper would benefit from reducing the number of hypotheses explored and expanding on one or two. My suggestion is that the story would greatly benefit from a deeper study on the role of IL-1 β in the IL-22 effect.

In addition, there are several points that should be addressed by the authors:

Major issues:

- 1.- The title/abstract highlight both IL-23 and IL-22, whereas the results focus primarily on IL-22. The title/abstract should reflect this.**
- 2.- The IL-23 axis of the treatment effect is not explored and rather understated – high IL-23 ES also correlates with poor response (albeit less so than for IL-22 ES), what are possible explanations for this? In addition, the comparison between the IL-23 and IL-22 ES stratification is interpreted as evidence of IL-22 being more critical in response to ustekinumab – could this difference not be attributed to the difference in signature accuracy, contribution of immune cells vs epithelial cells in colonic biopsy transcriptome or the difference in ES distribution in patients?**
- 3.- The signature genes used for the enrichment scores should be specified in order for further studies to benefit from these results**
- 4.- The interpretation of GSVA ES would be aided by some sort of explanation or illustration of what a high/low ES means (e.g. that the score is relative to the sample population, not an absolute value)**
- 5.- There is lack of consistency on the chemokines analyzed across the study. In Fig 6C it is only Cxcl5, in 6D it is only Cxcl1. This should be supplemented, justified or discussed in order to strengthen conclusion**
- 6.- The conclusion that CXCR2 deletion attenuates disease is quite strong in relation to the data (only reduced colon mass, whereas IL-22 results also included colitis score)**
- 7.- It is unclear whether the results of the co-culture results reflect only colonic organoids or total co-culture (i.e. could ILCs be expressing the measured chemokines?)**
- 8.- The author failed to discuss important studies reporting novel stratifications methodologies and results (Smillie C, Cell, 2019; Czarnewski, Nat Comm, 2019). In particular, it would be interesting to investigate if low or high IL22 ES correspond to either UC1/UC2 patients (unbiased molecular stratification from Czarnewski et al (2019)).**
- 9.- Figure 6A does not contribute to the rationale of selecting STAT3 and MAP3K8 as genes of interest (it would suffice to state that they are altered after IL-22 and that they**

are part of the recruitment of phagocytes pathway, as this is the only relevant info to be had from the figure). In addition, the red boxes hide genes, presumably STAT3 and MAP3K8?

10.- Please address the following issues regarding statistics/bioinformatics:

- The choices of bioinformatical results to display are not explained/justified, and it is therefore unclear whether the reader is being shown the strongest results or simply those which the authors found to be of interest, reducing the reliability of the results. For example, Figs 1C, 1F and 2A all represent similar analyses but are produced and illustrated in different ways. Also, the choice of GSVA makes sense for comparing IL-22 ES vs IL-23 ES but is not necessary for comparing treatment response. Does GSEA give similar results to GSVA in these cases?
- Several experiments are missing sample size and number of experiments. Please include the number of experiments in the fig. legends.
- The importance of IL-1 β compared to IL-23 is not statistically motivated in Fig 2D: It appears to be based on the increased difference in expression which is biased by the basal expression levels. The y-axis scale should be adjusted in each graph, and the correlation coefficient should be given, preferably with p-values
- DE models include "covariates accounting for [...] sample similarities detected by PCA and hierarchical clustering" – what are these covariates?
- Any type of statistical test (or even sample distribution) is missing in Fig 3D+E, Fig 4F, Fig 5H, Fig 6H, and thus the interpretation from these figures is highly limited
- Distribution of stratified ES scores in the sample population should be stated, as Fig 1G+H suggests very different numbers of patients in each group
- In order to compare stratified patients' response to placebo (line 170), the ES profile of the placebo group should be compared to that of the full treated population to ensure that this result is not biased.

Minor issues

- 1.- Some panels are lacking information on statistical tests: In Figs 5D+E, 6E, S8A it is left to the reader to interpret which groups have been tested against which.
- 2.- Several figures (2C, 4E, S4D+E, S5B) have elements (shapes, lines, colors) that are not explained. On the other hand, the legend of Fig 1F mentions green and red genes, but all genes are green.
- 3.- Improving/adding more schematics/icons and titles would improve understanding of figures. It would be helpful to indicate which experiments were performed in humans/mice
- 4.- Check and correct gene nomenclatures. For example, in line 633, mouse genes are not italicized, while in the graph legend of figure 5B "IL22-/-" should be "Il22-/-", as it is a mouse gene.
- 5.- Correct reference in Line 487.
- 6.- Temperature symbols should be corrected. (For example, in Line 677 it should be "37°C" instead of "370oC").
- 7.- References 17 and 28 are the same.
- 8.- Legends would overall benefit from adding more detail on experiments, notably in Figure 2.
- 9.- Resolution is low on several figures, some to the point where analysis is difficult

Reviewer #2 (Remarks to the Author):

In their paper entitled "The interleukin (IL)-12/IL-22 regulated transcriptome predicts treatment response and identifies mechanisms of treatment resistance in ulcerative colitis" Pavlidis and colleagues analyzed the transcriptional impact of IL-23 and IL-22 dependent signaling on clinical outcomes of patients with ulcerative colitis. Briefly, they describe transcriptional networks following treatment of lamina propria mononuclear

cells with IL-23 and colonic organoids with IL-22. These transcriptional networks were enriched in colonic biopsies of patients with ulcerative colitis (UC) enrolled into the phase III UNIFI trial assessing the efficacy of Ustekinumab vs. placebo. Surprisingly, enrichment of IL-22 negatively correlated with clinical outcomes following Ustekinumab therapy. The authors propose that this unexpected result could be explained by increased IL-1b signaling as well as activation of neutrophil-related chemokines in a STAT3-dependent manner, which may in turn promote IL-22. They provide evidence for comparable phenotypes in mouse models of inflammatory bowel disease, claiming that IL-22 may be a functionally important regulator of neutrophil recruitment in UC.

This is an interesting manuscript which combines data obtained in cell culture and animal models as well as clinical trials (i.e. the large and clinically important dataset from the UNIFI trial). Moreover, enrichment of the IL-22 signature was confirmed in a) organoids, b) human UC samples, c) Mouse models of IBD D) publicly available cohorts. These findings point towards neutrophil recruitment to the colon as important pathogenic factor driving UC pathogenesis.

However, the manuscript is sometimes a bit confusing and difficult to read in its present form. This is particularly due to the fact that the Figure legends do not provide sufficient information on what is actually shown and on experimental details. The reader repeatedly has to refer to the results and methods sections in order to get an idea on what he/she is supposed to see (e.g. Fig. 2: what was measured? in which material? what statistics and methodology was used in detail? Similarly, Figure 4 mixes data obtained from human and mouse colonoid cultures, mouse models of colitis as well as patient samples). This is rather difficult to follow, at least to me, and presentation of the data should be substantially clarified.

In addition, several issues should be addressed in order to further improve the quality of the manuscript:

1. In Fig. 1K/L, the authors stratify clinical outcomes following induction therapy with Ustekinumab according to the IL-22 enrichment score at baseline, i.e. before treatment. The observation that enrichment of the IL-22 signature negatively correlates with clinical outcomes is surprising. What is the precise sample size of each subgroup? How does stratification look like in the placebo group? More thorough analyses of prediction (sensitivity/specificity analyses etc.) should be provided if the authors claim that their signature may be used as a prognostic marker.

2. The authors should address the question as to whether the correlation with the IL-22 signature and non-response to therapy is specific for Ustekinumab. Can the signature applied to other datasets including patients treated with Anti-TNF and other drugs? Is there a correlation between the IL-22 signature and previous failure of anti-TNF therapy in the UNIFI cohort of patients?

3. IL-22 may have protective or deleterious functions in IBD. IBD pathogenesis and responsiveness to anti-TNF therapy has been linked with expression of IL-22 binding protein. Therefore the authors should address IL-22bp expression. Moreover, it would be interesting to see whether the observed IL-22 mediated signaling is concentration dependent.

4. If the authors have access to RNA data of the UNIFI trial programme, they should perform longitudinal analyses of the IL-23 and IL-22 transcriptional enrichment sets between week 0, week 8 and week 44 on a single patient level. Does it change over time/with disease course?

5. It is inappropriate to show a copy/paste photograph of colonic organoids in Fig. 1D. In contrast, it would be informative to know whether incubation with IL-22 had an effect on organoid cell morphology and cell fate.

6. Figure legends need to be specified (see above, this is particularly important for Fig. 2). In addition, quality of the Figures should be improved including readability,

alignments etc. If switching between models for data confirmation, add a header in the figure itself (e.g. Fig. 4). In addition, dosages should be indicated in the Figure legends and not only in the Methods section (e.g. for IL-23/IL-22 in Fig. 1). Moreover, specific numbers have to be indicated for each experiment. E.g. in Figure 1: LPMCs were isolated from how many patients? Some of the cited material is also missing (e.g. Table line 651; the cited TNF data (line 243) is not included in the referenced figures). The overlap of transcripts between the pathways cited in Fig. 3C is a bit weak: e.g. only 4 IL-17A transcripts were identified, which makes an analysis of overlap a bit difficult.

7). Details of the statistical analyses used for each experiment need to be given in the Figure legends. E.g. for Figs. 2D, 1K/L).

8) The authors have recently proposed that that IL-22 promotes coordinates ER stress responses in colonic organoids and patients with Crohn's (Powell et al., Gut 2020). How does this mechanism apply to the UC data reported here and vice versa?

9). In figure 4E, what kind of mice were used as wt controls? Apparently, mouse strains with BALBc as well as C57BL/6 background have been used. Therefore, suitable control animals with identical genetic background have to be used (preferentially littermate controls). Please also indicate the number of animals used for each experiment.

Reviewer #3 (Remarks to the Author):

Review of "The interleukin (IL)-23/IL22 regulated transcriptome predicts treatment response and identifies mechanisms of treatment resistance in ulcerative colitis"

This study evaluates the contribution of IL22 to the pathogenesis of UC. It is well written and supported by a large amount of scientifically rigorous data. The most exciting advancement is that the authors potential discovery that the IL22 RNA expression profile (derived from their organoids) is predictive of ustekinumab non-response when applied to patient biopsies. This finding could potentially be used to guide therapy decisions in the future, it's a relatively novel concept and could have wide reaching implications for using RNA expression profiles to understand disease.

The authors also demonstrate:

- 1) Characteristic changes in gene expression in vitro in colonic UC patient mononuclear cells after treatment with IL23 via RNA-seq
- 2) Changes in gene expression in vitro when colonic organoids are treated with rhIL22 via RNA-seq
- 3) Detailed characterization of possible up- and down-stream IL22 networks/pathways using in depth in silico analysis
- 4) IL22 regulation of neutrophil chemo-attractant expression
- 5) They also demonstrate in detail that IL22 is an important component of the TRUC murine model of colitis, through neutrophil recruitment
- 6) They confirm some IL22 mediated signalling mechanisms in murine epithelial cell specific Stat3 conditional knockout models

There are multiple IL22-related pathways and mechanisms explored in this study. Overall, this data is quite interesting and novel, but there is a heavy reliance on RNA expression from colonic organoids, making drawing strong conclusions and justifying the rationale for their in vitro studies challenging. They do use a murine colitis model elegantly to focus on neutrophil chemoattraction as an important part of IL22 signalling, but the connection to humans at the protein (or tissue or organism) level is lacking.

Major Suggestions

Results

I do not think the authors demonstrate that they can reliably differentiate patients who will respond to anti-IL12p40 therapy. The response percentages as quoted are not remarkable and overall low (26% in clinical remission) albeit double. I would not use this drug over anti-TNF based on these findings or low IL22.

In the UNIFI trial published in the NEJM paper the overall response rates are better than the response outlined here? Is that correct?

Key methodological issues make the results difficult to interpret.

Is there a table somewhere that defines the patient population? Age, sex, severity of disease, treatment. Is there any reason to think that these factors would not influence the subsequent results? How are these variables dealt with? Where the biopsies taken from the same colonic region of every patient - how many patients and bx per patient? Is there multiple samples from the same person showing that the results are reproducible?

This information is critical for non-drug trial patients studied here.

Another, perhaps stronger approach, would be to examine patients who failed anti-TNF therapy.

Define active UC in the - lots of variation here?

Line 107 - what are you comparing? Exposed vs non-exposed with UC? Is this the same response expected/shown in healthy controls? what is a normal response?

Where the organoids derived from (same region of colon) the same patient as the LPMC patients? How many?

Since IL23 effects expression of a broad range of important mediators (as the authors point out) and not just IL22, it seems plausible that IL23 may upregulate IL-22 expression indirectly through an intermediate mediator. When suggesting a multi-step molecular mechanism, it will be important to show downstream functional changes or at least protein-level data that IL23 induces IL22 function or protein level changes. This is especially true when the authors later note that patients with high IL22 enrichment scores had high expression of IL1B, another possible mediator in this mechanism.

Figure 1

The authors justify the further experiments in the manuscript on the concept that IL23 induces IL22 protein expression. Could cLMP supernatants be used to induce a similar RNA-seq profile as shown in Fig 1E? Can this effect be abrogated by IL23 blockade (which would provide stronger support for the proposed mechanism)? It will be important to confirm mechanistically how IL23 signals directly to IL22, given that this theoretical connection to ustekinumab therapy efficacy in patients is the most exciting finding in this study.

What is the rationale for the stratification levels (<0 , $0-0.6$ and >0.5) chosen in Fig 1K and 1L? Statistical rationale for these levels and n-values for each group in the Fig 1K and 1L plots will be important to strengthen this important finding.

Figure 2

Figure 2D – the differences in the second to fourth panels is stated as “substantial” but I could not find a statistical test used to verify this difference; a statistical test outcome (p-value or CI, etc) should be stated directly on the plots and again the justification for the stratification levels (<0 , $0-0.6$ and >0.5) is not clear but important if the IL22

expression profile could one day be used to guide the decision to treat patients with ustekinumab or not.

Figure 3

Neutrophil chemoattractant transcript expression appears to be regulated by IL22 and increased in active UC however this function is not confirmed by functional assay (such as neutrophil migration assay) or by showing altered protein expression – could the biopsies from the UNIFI study be immunostained or otherwise analyzed for neutrophil chemoattractant levels or even for neutrophil counts themselves (that could then be stratified by IL22 expression profile) to confirm a functional connection between IL22 and disease?

The authors should justify more clearly why they chose to pursue the IL17A/IL22 connection when the Th17 pathway is the 7th pathway listed in Figure 3A and also appears to have the lowest p-value of the pathways listed. That said, the Th17 pathway is a completely justifiable avenue to pursue based on literature that shows the well-described connection between IL17A and IL22 functions, especially in terms of chemoattraction through CXC chemokines (Valeri et al. *Pathogens and Disease*, 74: 9, 2016 and Liang et al. *J Exp Med*. 2006;203(10):2271-2279.).

Discussion

The discussion of anti-TNF in UC is out of date (references) and not correct with better treatment approaches including dosing and timing.

A graphical representation the proposed pathway leading from IL23 through IL22 to neutrophil tissue recruitment or unifying the experimental process (RNA expression profiles to murine colitis model) would help to bring together the many interesting experiments/findings in this study.

Minor Suggestions

Abstract

Line 31 “mostly” choice of word could be more objective, rewording sentence to be clearer would help.

The statement about IL22 being functionally and pathologically important in the recruitment of CXCR2+ neutrophils in colitis is not clearly demonstrated in this paper. I would suggest rewording this statement (see comment below about Figure 5).

Introduction

Sentence 51-52 is not really correct (no reference)

Intro is lacking key references for pathogenesis of UC.

Odd that no mention of anti-TNF drugs that have worked very well - would tone down the clinical description of how there is urgent need for treatment when anti-TNF works in most situations.

The first paragraph is very general and could focus more specifically on how available therapies are only loosely based on disease mechanism, leading to treatment failure
The trial mentioned on line 87 is not just commenced but completed as of Oct 2020 (although not yet published that I could find).

Results

Line 167 there is an extra “vs” that can be removed

Figure 1B: the authors state on line 110 that IL22 was the most significantly

upregulated gene but in the figure IFNG appears to have a lower FDR, does this not mean that the IFNG is the most significantly upregulated?

Line 216 should read "IL12B" not "IL123B"

The references at the end of the sentence on line 213/214 should be distributed throughout the sentence not grouped at the end, they were challenging to verify

Suppl Figure 6C: the network 2 and 3 outlines overlap but there is in fact no connection between these networks. The networks 1/2 and 3/4 should be separated from one another and discussed separately.

Suppl Figure 7C: the dose of IL22 used in the left panel should be clearly stated so it can be cross referenced with the right panel; likewise, the length of time of IL22 exposure in the right panel should be clearly stated so it can be cross referenced with the left panel

The sentence on line 324-325 should have a reference (or direct the reader to data) to justify this statement

Figure 5H: it is not clear how this panel fits into the rest of the results, the anti-neutrophil antibody would be expected to decrease neutrophil chemotaxis, the connection to IL22 dependent ligands is weak given that multiple other neutrophil chemokines also act through this receptor

Figure 6A: the red boxes I presume highlight STAT3 and MAP3K8 but they are covering these words in the pdf

Discussion

The expression profiles and network analysis repeatedly identify cell movement-related transcript expression changes as a result of IL22 stimulation, but this is not addressed in the authors' proposed mechanism or further elucidated with experiments, the relevance of this finding should at least be discussed.

The second paragraph focused mostly on IL23, a minor part of the data in this study. I would consider trimming this paragraph or moving the third paragraph which discusses the most important discovery in this study (the IL22 expression profile predictive of ustekinumab non-response) above the second paragraph.

Methods

Overall, the order of the subsections is hard to follow making cross referencing the results with the experimental methods challenging. I would suggest listing all the experimental models and patient recruitment first then presenting the specific experiments in order.

The status of the tissue where the mucosal biopsies were obtained is important to mention (were all biopsies taken from macroscopically inflamed areas?)

Line 609: it should be stated if the IBD patients were on therapy at the time of sampling and what therapy they were on

Line 654: 'HT' should be defined. The TRUC microbiota used should be described.

Point-by-point discussion of the reviewer comments

Firstly, we would also like to extend our gratitude to the reviewers. In addressing their comments, we believe that our manuscript is now much improved.

REVIEWER COMMENTS

Reviewer #1 (Remarks to the Author):

In this article, Pavlidis et al established a novel and well thought out method for stratifying patients based on the transcriptomic signature triggered by IL-23 and IL-22. First, the authors obtained these gene signatures using either IL23-stimulated LP mononuclear cells or IL22-stimulated human colonic organoids. Using Gene Set Variation Analysis (GSVA, a sample-specific version of the well-known Gene Set Enrichment Analysis (GSEA)), they evaluated the enrichment of these signature genes in ulcerative colitis patients and healthy controls and correlate the enrichment score (ES) with response or non-response to ustekinumab. Counterintuitively, perhaps in their most important finding, they found that a high IL-22 ES correlated with poor response to ustekinumab. In order to gain mechanistical insights, the authors reproduced the high IL-22 ES in 6 separate mouse models of UC. The authors also investigated possible upstream and downstream mediators of this response and presented

several pieces of evidence implicating the role of neutrophil recruitment in the detrimental effects of IL-22. The study holds great potential for improving treatment in ulcerative colitis, both through understanding of pathological events and opening up avenues for personalized treatment through stratification of patients based on colonic biopsies. However, the story overall is disjointed: The results indicating the IL22 downstream effects (eg the role of chemokines, role of STAT3) is somehow disconnected to the most interesting finding which is the identification of IL-17A as a potential cause of IL-22 ES and lack of responsiveness to ustekinumab. In particular, the relevance of IL17A is not clear and rather muddles the message of the paper. The paper would benefit from reducing the number of hypotheses explored and expanding on one or two. My suggestion is that the story would greatly benefit from a deeper study on the role of IL-17A in the IL-22 effect.

In addition, there are several points that should be addressed by the authors:

Major issues:

1.- The title/abstract highlight both IL-23 and IL-22, whereas the results focus primarily on IL-22. The title/abstract should reflect this.

Response: We thank the reviewer for their comments. We have made the suggested correction in both title and abstract. Taking on board this comment, and the sentiment expressed from all three reviewers, we agree that focussing on a more refined set of hypotheses would improve the messaging. Accordingly, we have focussed on the IL22/neutrophil axis, which is now the main message of the paper.

2.- The IL-23 axis of the treatment effect is not explored and rather understated – high IL-23 ES also correlates with poor response (albeit less so than for IL-22 ES), what are possible explanations for this? In addition, the comparison between the IL-23 and IL-22 ES

stratification is interpreted as evidence of IL-22 being more critical in response to ustekinumab – could this difference not be attributed to the difference in signature accuracy, contribution of immune cells vs epithelial cells in colonic biopsy transcriptome or the difference in ES distribution in patients?

Response: As requested, the IL23 work has now been removed, with emphasis now placed on the IL22/chemokine/neutrophil axis.

3.- The signature genes used for the enrichment scores should be specified in order for further studies to benefit from these results

Response: All relevant files and data as will be uploaded as supplementary files and all transcriptomic data will be reposted online. We have included a spreadsheet with all gene lists used for this experiment (R1.3_Gene_Lists), which will be a supplemental file.

4.- The interpretation of GSVA ES would be aided by some sort of explanation or illustration of what a high/low ES means (e.g. that the score is relative to the sample population, not an absolute value)

Response: Thank you for raising this point. We have now added a description that can guide the interpretation of enrichment scores using GSVA when the method is described for the first time in results: The score varies between +1 (upregulated) to -1 (downregulated) and depends on the distribution of gene expression across the samples tested.

5.- There is lack of consistency on the chemokines analyzed across the study. In Fig 6C it is only Cxcl5, in 6D it is only Cxcl1. This should be supplemented, justified or discussed in order to strengthen conclusion

Response: Thank you for your comment. We have now updated figure 6 for consistency and added new data in order to strengthen the conclusion. Please see point 9 below.

6.- The conclusion that CXCR2 deletion attenuates disease is quite strong in relation to the data (only reduced colon mass, whereas IL-22 results also included colitis score)

Response: we have now included new experimental data as requested by the reviewer. In support of our hypothesis, anti-CXCR2 treatment also significantly attenuated colitis scores.

7.- It is unclear whether the results of the co-culture results reflect only colonic organoids or total co-culture (i.e. could ILCs be expressing the measured chemokines?)

Response: In a previous experiment (unpublished data) we purified (FACS sorted) ILC3 from TRUC mice and stimulated them under optimal conditions (IL23 and IL1 β). Transcriptomic analysis with microarray identified Il22 as the most highly expressed transcript (FC:9, $p=1.25 \times 10^{-6}$), and importantly, Cxcl5 and Cxcl1 were not differentially expressed. Hence, we believe that the measured expression of chemokines represents colonoid origin. The DEG list can be provided to the reviewer if needed. For further clarity of this issue, we have interrogated single cell RNA-seq data from Smillie (Cell 2019). Importantly, in ILCs there was

minimal expression of CXCL1 and CXCL5 was completely undetectable (including in ILCs from inflamed colon, see below).

A) tSNE analysis showing the immune cell compartments in the Smillie et al dataset. B) tSNE plot showing CXCL1 and CXCL5 expression across the different immune populations. C) Expression levels of CXCL1 and CXCL5 in ILCs.

8.- The author failed to discuss important studies reporting novel stratifications methodologies and results (Smillie C, Cell, 2019; Czarnewski, Nat Comm, 2019). In particular, it would be interesting to investigate if low or high IL22 ES correspond to either UC1/UC2 patients (unbiased molecular stratification from Czarnewski et al (2019)).

Response: We have now updated the discussion to account for the reviewer's helpful comment. Following on from their comment we have used the 57 genes discriminating UC1 and UC2 cohorts from Czarnewski et al. to stratify the UNIFI cohort. One third of these genes

were upregulated by IL22 (over enrichment 10.16 fold, $p=9.360110470840996e-18$, hypergeometric test). We found a positive correlation between the enrichment score for the UC1 gene set and the IL22 regulated program and we also comment on the role of IL22 program activation in predicting response to other biologics, addressing similar comments from the other reviewers. Importantly, these new data potentially provide valuable mechanistic insights into the upstream regulation of the UC1UC2 signature, implicating IL22 as a potentially important regulator of the UC1UC2 transcriptional programme.

9.- Figure 6A does not contribute to the rationale of selecting STAT3 and MAP3K8 as genes of interest (it would suffice to state that they are altered after IL-22 and that they are part of the recruitment of phagocytes pathway, as this is the only relevant info to be had from the figure). In addition, the red boxes hide genes, presumably STAT3 and MAP3K8?

Response: Following on from your helpful feedback, we have now removed panel 6A. We are providing expression data on both Cxcl1 and Cxcl5 for the Stat3^{ΔIEE} and the Map3k8^{-/-} mouse experiment. We removed the data on pharmacological inhibition in mouse organoids and replaced it with an experiment of human colonic organoids (biological replicates, n=3) treated with IL22 and/or IL22+ tofacitinib, a JAK1/3 inhibitor licenced for use in UC. We believe that these new data provide compelling support on the IL22/JAK/STAT/chemokine axis in the human setting.

10.- Please address the following issues regarding statistics/bioinformatics:

- The choices of bioinformatical results to display are not explained/justified, and it is therefore unclear whether the reader is being shown the strongest results or simply those which the authors found to be of interest, reducing the reliability of the results. For example, Figs 1C, 1F and 2A all represent similar analyses but are produced and illustrated in different ways. Also, the choice of GSVA makes sense for comparing IL-22 ES vs IL-23 ES but is not necessary for comparing treatment response. Does GSEA give similar results to GSVA in these cases?

Response. Thank you for this feedback. We have updated figs 1f with a depiction of the top 10 pathways enriched in the IL22 regulated transcriptional programme using the R package clusterProfiler (hallmark gene sets, KEGG, Reactome pathway annotation from the MSigDB database). GSEA and GSVA give very similar results in this context, you can see below a correlation between the two methods for the IL22 enrichment score (UNIFI trial, n=550).

- Several experiments are missing sample size and number of experiments. Please include the number of experiments in the fig. legends.

Response: Thank you for your comment, amendments made across the manuscript.

- The importance of IL-1B compared to IL-23 is not statistically motivated in Fig 2D: It appears to be based on the increased difference in expression which is biased by the basal expression levels. The y-axis scale should be adjusted in each graph, and the correlation coefficient should be given, preferably with p-values

Response: We have changed the figure as per reviewer's comment and provided additional analysis in the form of a correlation matrix.

- DE models include "covariates accounting for [...] sample similarities detected by PCA and hierarchical clustering" – what are these covariates?

Response: We have now included the model we used for DEG analysis in the methods

$$\mu_i = \beta_0 + \beta_{t[i]} + \beta_{p[j]} + \beta_{c[k]}$$

$$y_i \sim \text{NegativeBinomial}(\exp(\mu_i), \phi)$$

$$\beta_{t[i]} \sim \text{Normal}(0, \sigma_1)$$

$$\beta_{p[j]} \sim \text{Normal}(0, \sigma_2)$$

$$\beta_{c[k]} \sim \text{Normal}(0, \sigma_3)$$

where :

t = number of groups in Treatment
 p = number of subjects
 c = number of clusters
 i = number of observations
 y = gene expression count
 ϕ = overdispersion parameter
 σ_1 = between Treatment standard deviation
 σ_2 = between subject standard deviation
 σ_3 = between cluster standard deviation

- Any type of statistical test (or even sample distribution) is missing in Fig 3D+E, Fig 4F, Fig 5H, Fig 6H, and thus the interpretation from these figures is highly limited

Response: Figures 3D, 3E, 4F are all snapshots of DEG analysis- we will upload all DEG lists as supplemental material. Figure 6H has now been removed. Following on from reviewer 3 comments we have now removed figure 5H.

- Distribution of stratified ES scores in the sample population should be stated, as Fig 1G+H suggests very different numbers of patients in each group

Response: Another way to show these results is the following which we believe addresses the reviewer's comment.

These bar graphs depict the number of patients per group treated with ustekinumab in the induction phase of the UNIFI study. Outcomes from week 8 are reported. We have also provided the % for each group and the p value for the x² statistic for each outcome. We will include these graphs as supplementary data.

- In order to compare stratified patients' response to placebo (line 170), the ES profile of the placebo group should be compared to that of the full treated population to ensure that this result is not biased.

Response: Thank you for raising this issue. There is no statistical difference in the distribution of the IL22ES between the two groups as shown below:

In view of multiple supplementary figures we did not include these data but will happily include if reviewers/editors felt this was necessary.

Minor issues

1.- Some panels are lacking information on statistical tests: In Figs 5D+E,6E, S8A it is left to the reader to interpret which groups have been tested against which.

Corrected, thank you.

2.- Several figures (2C, 4E, S4D+E, S5B) have elements (shapes, lines, colors) that are not explained. On the other hand, the legend of Fig 1F mentions green and red genes, but all genes are green.

Corrected, thank you.

3.- Improving/adding more schematics/icons and titles would improve understanding of figures. It would be helpful to indicate which experiments were performed in humans/mice

Additional information added as per suggestion.

4.- Check and correct gene nomenclatures. For example, in line 633, mouse genes are not italicized, while in the graph legend of figure 5B "IL22-/-" should be "Il22-/-", as it is a mouse gene.

5.- Correct reference in Line 487.

6.- Temperature symbols should be corrected. (For example, in Line 677 it should be "37°C" instead of "370oC").

7.- References 17 and 28 are the same.

8.- Legends would overall benefit from adding more detail on experiments, notably in Figure 2.

9.- Resolution is low on several figures, some to the point where analysis is difficult

Corrected, thank you.

Reviewer #2 (Remarks to the Author):

In their paper entitled "The interleukin (IL)-12/IL-22 regulated transcriptome predicts treatment response and identifies mechanisms of treatment resistance in ulcerative colitis" Pavlidis and colleagues analyzed the transcriptional impact of IL-23 and IL-22 dependent signaling on clinical outcomes of patients with ulcerative colitis. Briefly, they describe transcriptional networks following treatment of lamina propria mononuclear cells with IL-23 and colonic organoids with IL-22. These transcriptional networks were enriched in colonic biopsies of patients with ulcerative colitis (UC) enrolled into the phase III UNIFI trial assessing the efficacy of Ustekinumab vs. placebo. Surprisingly, enrichment of IL-22 negatively correlated with clinical outcomes following Ustekinumab therapy. The authors propose that this unexpected result could be explained by increased IL-1b signaling as well as activation of neutrophil-related chemokines in a STAT3-dependent manner, which may in turn promote IL-22. They provide evidence for comparable phenotypes in mouse models of inflammatory bowel disease, claiming that IL-22 may be a functionally important regulator of neutrophil recruitment in UC.

This is an interesting manuscript which combines data obtained in cell culture and animal models as well as clinical trials (i.e. the large and clinically important dataset from the UNIFI trial). Moreover, enrichment of the IL-22 signature was confirmed in a) organoids, b) human UC samples, c) Mouse models of IBD D) publicly available cohorts. These findings point towards neutrophil recruitment to the colon as important pathogenic factor driving UC pathogenesis.

However, the manuscript is sometimes a bit confusing and difficult to read in its present form. This is particularly due to the fact that the Figure legends do not provide sufficient information on what is actually shown and on experimental details. The reader repeatedly has to refer to the results and methods sections in order to get an idea on what he/she is supposed to see (e.g. Fig. 2: what was measured? in which material? what statistics and methodology was used in detail? Similarly, Figure 4 mixes data obtained from human and mouse colonoid cultures, mouse models of colitis as well as patient samples). This is rather difficult to follow, at least to me, and presentation of the data should be substantially clarified.

Response: We thank the reviewer for these helpful comments, which we have now implemented to improve the clarity of the study.

In addition, several issues should be addressed in order to further improve the quality of the manuscript:

1. In Fig. 1K/L, the authors stratify clinical outcomes following induction therapy with Ustekinumab according to the IL-22 enrichment score at baseline, i.e. before treatment. The observation that enrichment of the IL-22 signature negatively correlates with clinical outcomes is surprising. What is the precise sample size of each subgroup? How does stratification look like in the placebo group? More thorough analyses of prediction (sensitivity/specificity analyses etc.) should be provided if the authors claim that their signature may be used as a prognostic marker.

We are providing additional figures that include number of patients for each subgroup (addressed above, for a similar comment by reviewer #1). Below you can see the stratification of the placebo treated patients as per the IL22 enrichment score.

While this is not a primarily biomarker discovery/evaluation oriented manuscript we agree with the reviewer that including the ROC analysis can be helpful to the reader and can be included as supplementary data.

2. The authors should address the question as to whether the correlation with the IL-22 signature and non-response to therapy is specific for Ustekinumab. Can the signature applied to other datasets including patients treated with Anti-TNF and other drugs? Is there a correlation between the IL-22 signature and previous failure of anti-TNF therapy in the UNIFI cohort of patients?

This is an important point. In the UNIFI cohort, patients with previous failure to antiTNF therapy are found to have a higher IL22 enrichment score. When we analysed previously repositied datasets (GSE23597, GSE16879, GSE92457, GSE73661) reporting on response to infliximab, golimumab or vedolizumab IL22 ES appeared to be higher in infliximab non responders only in one study (GSE16879). These new data have now been included in the manuscript.

GEO	GSE23597	GSE16879	GSE92415	GSE73661
n	45	24	59	60
drug	infliximab	infliximab	golimumab	vedolizumab
time point	wk8	wk8	wk6	wk4-6
endpoint	clinical response	mucosal healing	clinical response	mucosal healing

*(Mann Whitney test, *p<0.05)*

3. IL-22 may have protective or deleterious functions in IBD. IBD pathogenesis and responsiveness to anti-TNF therapy has been linked with expression of IL-22 binding protein.

Therefore the authors should address IL-22bp expression. Moreover, it would be interesting to see whether the observed IL-22 mediated signalling is concentration dependent.

Response: We have looked at the expression of IL22RA2 in the UNIFI dataset (UC patients only) in association to IL22 expression and the enrichment of the IL22 regulated programme. Expression of IL22RA2 does not correlate with either of the two ($r=0.05$, 95%CI[-0.03, 0.138], $p=0.22$ and $r= -0.017$, 95%CI[-0.10, -.6], $p=0.69$, Spearman r).

We have already provided data to address the point raised by the reviewer in regards to the concentration dependence of the IL-22 mediated signalling by reporting on the expression of the chemokines Cxcl1 and Cxcl5 in mouse colonoids. To reinforce the message we have repeated the experiment with human colonoids with similar results.

4. If the authors have access to RNA data of the UNIFI trial programme, they should perform longitudinal analyses of the IL-23 and IL-22 transcriptional enrichment sets between week 0, week 8 and week 44 on a single patient level. Does it change over time/with disease course?

Response: We agree that this would be an interesting question, but we only have access to W0 and W8 transcriptomic data (there were very few transcriptomic samples available in W44 patients). We observed a drop in IL22 ES (and IL23 ES) with ustekinumab therapy from wk0 to wk8 which was statistically significant in those who achieved mucosal healing. We have emphasized these findings.

5. It is inappropriate to show a copy/paste photograph of colonic organoids in Fig. 1D. In contrast, it would be informative to know whether incubation with IL-22 had an effect on organoid cell morphology and cell fate.

Response: We have amended the photograph using a representative photo of untreated and IL22 treated colonoids. Treatment with IL22 did not affect area or perimeter of human organoids from two donors (Mann Whitney test). We did not expect to see changes in this relatively short period of time. The intention of the experiment was to generate the transcriptional footprint of IL22 in human colonoids. The effects of IL22 on morphological changes in the epithelium in longer term cultures have been established¹. In view of multiple

supplementary figures we did not include these data but will happily include if reviewers/editors felt this was necessary.

6. Figure legends need to be specified (see above, this is particularly important for Fig. 2). In addition, quality of the Figures should be improved including readability, alignments etc. If switching between models for data confirmation, **add a header in the figure itself** (e.g. Fig. 4). In addition, dosages should be indicated in the Figure legends and not only in the Methods section (e.g. for IL-23/IL-22 in Fig. 1). Moreover, specific numbers have to be indicated for each experiment. E.g. in Figure 1: LPMCs were isolated from how many patients? Some of the cited material is also missing (e.g. Table line 651; the cited TNF data (line 243) is not included in the referenced figures). *Corrected, thank you. Line 243 refers to predicted pathways that were identified on pathway analysis.* The overlap of transcripts between the pathways cited in Fig. 3C is a bit weak: e.g. only 4 IL-17A transcripts were identified, which makes an analysis of overlap a bit difficult. *The numbers in fig3c refer to pathways rather than DEG as described in the legend.*

7). Details of the statistical analyses used for each experiment need to be given in the Figure legends. E.g. for Figs. 2D, 1K/L). *Corrected, thank you.*

8) The authors have recently proposed that that IL-22 promotes coordinates ER stress responses in colonic organoids and patients with Crohn's (Powell et al., Gut 2020). How does this mechanism apply to the UC data reported here and vice versa?

*Response: Using the UNIFI dataset (UC patients only, n=542) we find a positive correlation between the IL22 and ER stress enrichment scores (ES), $r = 0.59$, 95%CI[0.53,0.64], $p < 0.0001$. When we stratify ustekinumab treated patients by their IL22 ES we also see a positive association (Kruskal-Wallis test, $***p < 0.0001$, comparisons 0-0.5 vs <0, >0.5 vs <0 and >0.5 vs 0-0.5, all with $P < 0.0001$, see below). These data support our previous observations that IL22 is likely to be an inducer of ER stress in human colonic epithelial cells. The potential for a relationship between neutrophil recruitment and ER stress in the epithelium is potentially very interesting, and may even be causally related (it is possible that IL22 mediated induction of neutrophil recruitment could even drive/exacerbate ER stress. However, we believe that this question is beyond the scope of the current paper.*

9). In figure 4E, what kind of mice were used as wt controls? Apparently, mouse strains with BALBc as well as C57BL/6 background have been used. Therefore, suitable control animals with identical genetic background have to be used (preferentially littermate controls). Please also indicate the number of animals used for each experiment.

Response: We have amended the figure as per reviewer's comment showing the appropriate control next to the relevant colitis model. Littermates were used as appropriate controls.

Reviewer #3 (Remarks to the Author):

Review of "The interleukin (IL)-23/IL22 regulated transcriptome predicts treatment response and identifies mechanisms of treatment resistance in ulcerative colitis"

This study evaluates the contribution of IL22 to the pathogenesis of UC. It is well written and supported by a large amount of scientifically rigorous data. The most exciting advancement is that the authors potential discovery that the IL22 RNA expression profile (derived from their organoids) is predictive of ustekinumab non-response when applied to patient biopsies. This finding could potentially be used to guide therapy decisions in the future, it's a relatively novel concept and could have wide reaching implications for using RNA expression profiles to understand disease.

The authors also demonstrate:

- 1) Characteristic changes in gene expression in vitro in colonic UC patient mononuclear cells after treatment with IL23 via RNA-seq
- 2) Changes in gene expression in vitro when colonic organoids are treated with rhIL22 via RNA-seq
- 3) Detailed characterization of possible up- and down-stream IL22 networks/pathways using in depth in silico analysis
- 4) IL22 regulation of neutrophil chemo-attractant expression
- 5) They also demonstrate in detail that IL22 is an important component of the TRUC murine model of colitis, through neutrophil recruitment
- 6) They confirm some IL22 mediated signalling mechanisms in murine epithelial cell specific Stat3 conditional knockout models

There are multiple IL22-related pathways and mechanisms explored in this study. Overall, this data is quite interesting and novel, but there is a heavy reliance on RNA expression from

colonic organoids, making drawing strong conclusions and justifying the rationale for their in vitro studies challenging. They do use a murine colitis model elegantly to focus on neutrophil chemoattraction as an important part of IL22 signalling, but the connection to humans at the protein (or tissue or organism) level is lacking.

Response: We thank the reviewer for the recognition of our work and supportive comments. We have performed further work to address the points raised and in particular the last comment regarding the connection of IL22 signalling to neutrophil chemoattraction at tissue/organism level. Following all reviewers' comments, aiming to increase focus and readability of our manuscript we have now removed the description of the IL23 transcriptional program and focused on the IL22/neutrophil chemotaxis axis.

Major Suggestions

Results

I do not think the authors demonstrate that they can reliably differentiate patients who will respond to anti-IL12p40 therapy. The response percentages as quoted are not remarkable and overall low (26% in clinical remission) albeit double. I would not use this drug over anti-TNF based on these findings or low IL22.

Response: The reviewer raises an important point on the clinical applicability of the presented cytokine regulated signatures for stratifying patients. We agree with them that this is not a tool that can be used in clinical practice in its current form but as they rightly recognise 'potentially (it can) be used to guide therapy decisions in the future, it's a relatively novel concept and could have wide reaching implications for using RNA expression profiles to understand disease'. Even though we are not necessarily making a strong case about the biomarker aspect of the paper, it is a fair comment to say that doubling the response to ustekinumab based on a stratification tool such as IL22 enrichment score would be a major advance in the treatment of IBD (there is currently no such tool). Nevertheless, we have not made a major argument about this point, and instead have prioritized the mechanistic aspects of the study.

In the UNIFI trial published in the NEJM paper the overall response rates are better than the response outlined here? Is that correct?

Response rates are slightly different as only a subpopulation of the UNIFI trial cohort had biopsies processed for tissue transcriptional profiling.

Key methodological issues make the results difficult to interpret.

Is there a table somewhere that defines the patient population? Age, sex, severity of disease, treatment. Is there any reason to think that these factors would not influence the subsequent results? How are these variables dealt with? Where the biopsies taken from the same colonic region of every patient - how many patients and bx per patient? Is there multiple samples from the same person showing that the results are reproducible?

Response: We have added a demographics table describing the cohort included in this paper. In regards to demographics parameters affecting response to ustekinumab we would refer

the reviewer to the original publication in NEJM. All biopsies were taken from the same colonic region for every patient as per study protocol (15 to 20 cm from the anal verge). We have included 550 UC patients who participated in UNIFI and 18 healthy controls. Transcriptomic profiling of multiple samples from the same person were not part of the study protocol.

This information is critical for non-drug trial patients studied here.

Response: All patients reported in this manuscript participated in UNIFI.

Another, perhaps stronger approach, would be to examine patients who failed anti-TNF therapy.

Response: This is an important point (also raised by another reviewer). We have addressed this important point by looking at anti-TNF experienced patients who participated in UNIFI as well as repositing datasets. We have now included this new analysis as supplementary data.

Define active UC in the - lots of variation here?

Line 107 - what are you comparing? Exposed vs non-exposed with UC? Is this the same response expected/shown in healthy controls? what is a normal response?

Response: This part of the manuscript has now been removed.

Where the organoids derived from (same region of colon) the same patient as the LPMC patients? How many?

Response: The following paragraph is included in methods under section: 'human and mouse colonoids'

Human colonic crypts were isolated from serial colonic biopsies (x2 ascending colon, x2 transverse, x2 descending, x2 rectosigmoid) taken from four adult individuals (median age: 48, range[33,67], female:2), without past medical history or regular medication who attended for routine colonoscopy in view of abdominal symptoms without a diagnosis of IBD and did not have macroscopic or microscopic evidence of inflammation. All patients provided informed consent (NRES/IRAS id: 15/LO/1998).

Since IL23 effects expression of a broad range of important mediators (as the authors point out) and not just IL22, it seems plausible that IL23 may upregulate IL-22 expression indirectly through an intermediate mediator. When suggesting a multi-step molecular mechanism, it will be important to show downstream functional changes or at least protein-level data that IL23 induces IL22 function or protein level changes. This is especially true when the authors later note that patients with high IL22 enrichment scores had high expression of IL1B, another possible mediator in this mechanism.

Figure 1

The authors justify the further experiments in the manuscript on the concept that IL23 induces IL22 protein expression. Could cLMP supernatants be used to induce a similar RNA-seq profile as shown in Fig 1E? Can this effect be abrogated by IL23 blockade (which would provide stronger support for the proposed mechanism)? It will be important to confirm mechanistically how IL23 signals directly to IL22, given that this theoretical connection to ustekinumab therapy efficacy in patients is the most exciting finding in this study.

On the basis of comments from all reviewers we have now removed the IL23 work, and accordingly have not pursued mechanistically how IL23 signals directly. The role of IL23 and IL18 as upstream regulators of IL22 production is well described²⁻⁸.

For the referees interest we have included in this response a comprehensive network to highlight the likely mechanism.

We adapted the Virallink pipeline⁹ to connect IL23 to IL22 expression. Briefly, we generated a causal network by linking the receptors to IL22-regulating transcription factors using the Tied Diffusion Through Interacting Events (TieDIE) network diffusion approach¹⁰(figure1). We collected IL23 receptors using the OmniPathR R package (<https://github.com/saezlab/OmnipathR>) while IL22 regulating transcription factors were derived from the literature [Table X]. Directed and signed protein-protein interactions were downloaded from OmniPath¹¹ and contextualised by removing any interactions where either gene was not expressed in the IL23-treated LPMC samples. We filtered the causal networks by eliminating the potential ligands using the SignaLink database¹². A gene was considered expressed if its log2 expression value was above the mean minus 2 standard deviations of the expressed genes¹³. Gene IDs were converted using the 'AnnotationDbi' package in R. Network investigation, including centrality measures and functional analysis was carried out as described in Treveil et al⁹.

Transcription factor	Sign	Reference
RORC (RORgt)	activatory	14,15
STAT3	activatory	16
RUNX1	activatory	17
AHR	activatory	17
TBX21 (T-BET)	activatory	17
HIF1A	activatory	18
MAF (C-MAF)	inhibitory	19

We measured the betweenness centrality parameter for each protein in the network to compare their global importance in regulating the expression of IL-22. Results show that the

betweenness centrality differs among the three networks: in the IL23 - IL22 network the most central proteins are STAT1 and STAT3.

Functional analysis - using pathway data from Reactome and annotations from Gene Ontology (GO) database - revealed that among the top ten most significant functions, in the IL23 - IL22 network mainly the interleukin-related pathways are dominating (IL2-, IL3-, IL4-, IL5- and IL13-mediated signalling)

Figure 1: Effect of IL23 on IL22 gene expression. The colour of nodes (proteins) represents the betweenness value. Grey nodes represent the start (ligand) and end (IL22 gene) points of the network. Edge between nodes show the activatory or inhibitory connection. Figure was created in Cytoscape (12).

Supplementary Figure 1: Top ten overrepresented function in IL23 - IL22 network.

What is the rationale for the stratification levels (<0, 0-0.6 and >0.5) chosen in Fig 1K and 1L? Statistical rationale for these levels and n-values for each group in the Fig 1K and 1L plots will be important to strengthen this important finding.

Response: We have generated new figures that include patient numbers per stratified group as well as percentages of response per group (see response to reviewer 1). The cut off decision was based on our interpretation of the GSVA score and what we felt it represented at functional level (i.e. <0- programme not active, 0-0.5 some activation, >0.5 high activation). In addition, the limit for the upper quartile was at 0.428 so we felt that 0.5 would be a reasonable cut off.

Figure 2

Figure 2D – the differences in the second to fourth panels is stated as “substantial” but I could not find a statistical test used to verify this difference; a statistical test outcome (p-value or CI, etc) should be stated directly on the plots and again the justification for the stratification levels (<0, 0-0.6 and >0.5) is not clear but important if the IL22 expression profile could one day be used to guide the decision to treat patients with ustekinumab or not.

Response: We have updated the figure and legend with information on statistical tests used.

Figure 3

Neutrophil chemoattractant transcript expression appears to be regulated by IL22 and increased in active UC however this function is not confirmed by functional assay (such as neutrophil migration assay) or by showing altered protein expression – could the biopsies from the UNIFI study be immunostained or otherwise analyzed for neutrophil chemoattractant levels or even for neutrophil counts themselves (that could then be stratified by IL22 expression profile) to confirm a functional connection between IL22 and disease?

Response: This is an important point, and we have now addressed this question with new experimental data. We have performed new experiments comparing the IL22 ES with neutrophil counts in tissue (H&E staining) of the UC patients included in the UNIFI study. In support of our hypothesis, work we identified a positive correlation between neutrophil recruitment and activation (enrichment) of the IL22 transcriptional programme at week 0 (prior to starting therapy).

In silico analysis using CIBERSORT, a package that infers cellular composition from bulk transcriptomic data including microarrays, also revealed a positive correlation between IL22 ES and neutrophil presence (Spearman r 0.64, 95%CI [0.59, 0.69], $p < 0.0001$). We have included the Geboes neutrophil subscores in the main figure panel and included the CIBERSORT data as a supplementary figure.

The authors should justify more clearly why they chose to pursue the IL17A/IL22 connection when the Th17 pathway is the 7th pathway listed in Figure 3A and also appears to have the lowest p-value of the pathways listed. That said, the Th17 pathway is a completely justifiable avenue to pursue based on literature that shows the well-described connection between IL17A and IL22 functions, especially in terms of chemoattraction through CXC chemokines (Valeri et al. *Pathogens and Disease*, 74: 9, 2016 and Liang et al. *J Exp Med*. 2006;203(10):2271-2279.).

Response: Thank you for your point, we have amended the text. As suggested, the focus in IL17A was due to the fact that it has been traditionally seen as the cytokine associated with neutrophil chemoattraction.

Discussion

The discussion of anti-TNF in UC is out of date (references) and not correct with better treatment approaches including dosing and timing.

Response: We have updated with latest clinical trial data.

A graphical representation the proposed pathway leading from IL23 through IL22 to neutrophil tissue recruitment or unifying the experimental process (RNA expression profiles to murine colitis model) would help to bring together the many interesting experiments/findings in this study.

Response: We agree that this would be an excellent addition to the manuscript. We acknowledge that a graphical representation is an editorial decision, but would be very happy to prove one (or work with the graphics team to generate one). We envisage a figure as shown below.

Minor Suggestions

Abstract

Line 31 “mostly” choice of word could be more objective, rewording sentence to be clearer would help.

Corrected, thank you.

The statement about IL22 being functionally and pathologically important in the recruitment of CXCR2+ neutrophils in colitis is not clearly demonstrated in this paper. I would suggest rewording this statement (see comment below about Figure 5).

The addition of the histology scores and additional data to support fig5 we believe reinforce the message of our work linking IL22 to neutrophil recruitment and disease refractoriness to ustekinumab.

Introduction

Sentence 51-52 is not really correct (no reference)

Intro is lacking key references for pathogenesis of UC.

Odd that no mention of anti-TNF drugs that have worked very well - would tone down the clinical description of how there is urgent need for treatment when anti-TNF works in most situations.

The first paragraph is very general and could focus more specifically on how available therapies are only loosely based on disease mechanism, leading to treatment failure
The trial mentioned on line 87 is not just commenced but completed as of Oct 2020 (although not yet published that I could find).

We have addressed all above points by amending the introduction.

Results

Line 167 there is an extra “vs” that can be removed

Corrected, thank you

Figure 1B: the authors state on line 110 that IL22 was the most significantly upregulated gene but in the figure IFNG appears to have a lower FDR, does this not mean that the IFNG is the most significantly upregulated?

Line 216 should read “IL12B” not “IL123B”

Corrected, thank you

The references at the end of the sentence on line 213/214 should be distributed throughout the sentence not grouped at the end, they were challenging to verify

Changed as per comment.

Suppl Figure 6C: the network 2 and 3 outlines overlap but there is in fact no connection between these networks. The networks 1/2 and 3/4 should be separated from one another and discussed separately.

We have re-drawn the figure. We have provided additional information to the figure to assist with interpretation.

Suppl Figure 7C: the dose of IL22 used in the left panel should be clearly stated so it can be cross referenced with the right panel; likewise, the length of time of IL22 exposure in the right panel should be clearly stated so it can be cross referenced with the left panel

Additional information as requested has been added to the legend, confirming that the results are comparable.

The sentence on line 324-325 should have a reference (or direct the reader to data) to justify this statement

Reference added, thank you.

Figure 5H: it is not clear how this panel fits into the rest of the results, the anti-neutrophil antibody would be expected to decrease neutrophil chemotaxis, the connection to IL22 dependent ligands is weak given that multiple other neutrophil chemokines also act through this receptor

We have removed the panel in the amended version of our manuscript.

Figure 6A: the red boxes I presume highlight STAT3 and MAP3K8 but they are covering these words in the pdf

Figure has been removed following peer review comments re: redundancy.

Discussion

The expression profiles and network analysis repeatedly identify cell movement-related transcript expression changes as a result of IL22 stimulation, but this is not addressed in the authors' proposed mechanism or further elucidated with experiments, the relevance of this finding should at least be discussed.

The second paragraph focused mostly on IL23, a minor part of the data in this study. I would consider trimming this paragraph or moving the third paragraph which discusses the most important discovery in this study (the IL22 expression profile predictive of ustekinumab non-response) above the second paragraph.

We have amended the discussion taking the reviewer's comments under consideration.

Methods

Overall, the order of the subsections is hard to follow making cross referencing the results with the experimental methods challenging. I would suggest listing all the experimental models and patient recruitment first then presenting the specific experiments in order.

The status of the tissue where the mucosal biopsies were obtained is important to mention (were all biopsies taken from macroscopically inflamed areas?)

Line 609: it should be stated if the IBD patients were on therapy at the time of sampling and what therapy they were on

Line 654: 'HT' should be defined. The TRUC microbiota used should be described.

We have amended the methods to reflect the reviewer's comments as well as the peer review driven changes to the manuscript.

References

1. Lindemans, C.A., et al. Interleukin-22 promotes intestinal-stem-cell- mediated epithelial regeneration. *Nature*, 1-18 (2015).
2. Langrish, C.L., et al. IL-23 drives a pathogenic T cell population that induces autoimmune inflammation. *J Exp Med* 201, 233-240 (2005).
3. Kreymborg, K., et al. IL-22 is expressed by Th17 cells in an IL-23-dependent fashion, but not required for the development of autoimmune encephalomyelitis. *J Immunol* 179, 8098-8104 (2007).
4. Zheng, Y., et al. Interleukin-22, a T(H)17 cytokine, mediates IL-23-induced dermal inflammation and acanthosis. *Nature* 445, 648-651 (2007).

5. Ghoreschi, K., et al. Generation of pathogenic T(H)17 cells in the absence of TGF-beta signalling. *Nature* 467, 967-971 (2010).
6. El-Behi, M., et al. The encephalitogenicity of T(H)17 cells is dependent on IL-1- and IL-23-induced production of the cytokine GM-CSF. *Nat Immunol* 12, 568-575 (2011).
7. Rutz, S., Eidenschenk, C. & Ouyang, W. IL-22, not simply a Th17 cytokine. *Immunol Rev* 252, 116-132 (2013).
8. Gaffen, S.L., Jain, R., Garg, A.V. & Cua, D.J. The IL-23-IL-17 immune axis: from mechanisms to therapeutic testing. *Nat Rev Immunol* 14, 585-600 (2014).
9. Treveil A, Bohar B, Sudhakar P, Gul L, Csabai L, Olbei M, et al. ViralLink: An integrated workflow to investigate the effect of SARS-CoV-2 on intracellular signalling and regulatory pathways. *PLoS Comput Biol*. 2021 Feb 3;17(2):e1008685.
10. Paull EO, Carlin DE, Niepel M, Sorger PK, Haussler D, Stuart JM. Discovering causal pathways linking genomic events to transcriptional states using Tied Diffusion Through Interacting Events (TieDIE). *Bioinformatics*. 2013 Nov 1;29(21):2757-64.
11. Türei D, Valdeolivas A, Gul L, Palacio-Escat N, Klein M, Ivanova O, et al. Integrated intra- and intercellular signaling knowledge for multicellular omics analysis. *Mol Syst Biol*. 2021 Mar;17(3).
12. Fazekas D, Koltai M, Türei D, Módos D, Pálffy M, Dúl Z, et al. Signalink 2 - a signaling pathway resource with multi-layered regulatory networks. *BMC Syst Biol*. 2013 Jan 18;7:7.
13. Hart T, Komori HK, LaMere S, Podshivalova K, Salomon DR. Finding the active genes in deep RNA-seq gene expression studies. *BMC Genomics*. 2013 Nov 11;14:778.
14. Trifari S, Kaplan CD, Tran EH, Crellin NK, Spits H. Identification of a human helper T cell population that has abundant production of interleukin 22 and is distinct from T(H)-17, T(H)1 and T(H)2 cells. *Nat Immunol*. 2009 Aug;10(8):864-71.
15. Sekimata M, Yoshida D, Araki A, Asao H, Iseki K, Murakami-Sekimata A. Runx1 and ROR γ t Cooperate to Upregulate IL-22 Expression in Th Cells through Its Distal Enhancer. *J Immunol*. 2019 Jun 1;202(11):3198-210.
16. Backert I, Koralov SB, Wirtz S, Kitowski V, Billmeier U, Martini E, et al. STAT3 activation in Th17 and Th22 cells controls IL-22-mediated epithelial host defense during infectious colitis. *J Immunol*. 2014 Oct 1;193(7):3779-91.
17. Basu R, O'Quinn DB, Silberger DJ, Schoeb TR, Fouser L, Ouyang W, et al. Th22 cells are an important source of IL-22 for host protection against enteropathogenic bacteria. *Immunity*. 2012 Dec 14;37(6):1061-75.
18. Budda SA, Girton A, Henderson JG, Zenewicz LA. Transcription Factor HIF-1 α Controls Expression of the Cytokine IL-22 in CD4 T Cells. *J Immunol*. 2016 Oct 1;197(7):2646-52.
19. Rutz S, Noubade R, Eidenschenk C, Ota N, Zeng W, Zheng Y, et al. Transcription factor c-Maf mediates the TGF- β -dependent suppression of IL-22 production in T(H)17 cells. *Nat Immunol*. 2011 Oct 16;12(12):1238-45.

REVIEWER COMMENTS

Reviewer #1 (Remarks to the Author):

In this revised version, Pavlidis and colleagues followed the reviewer's suggestions and removed entirely the IL23 data, and add new data that reinforces the findings described in the first version, notably concerning the role of IL22 in neutrophil infiltration.

The revised version provides strong evidence that IL22, which is counterintuitively associated with detrimental effects in IBD patients, promotes the induction of chemokines associated with neutrophil migration. However, the data claiming that IL-22 transcript levels predict responsiveness to biological is rather less convincing. Thus, this reviewer considers that claims made by the authors are not supported by the data.

Major issues

1. There is still an issue in how the importance of IL22 in patients is being presented.
 - a. Firstly, several instances in the text (including the title) suggest that the IL22 ES has predictive power, but this is not supported by the data. Since the authors noted in the rebuttal that the score is not being proposed as a biomarker, it would suffice to change the title and text to reflect that the relation is associative, not predictive. (Title, abstract, pages 7 and 17, and S1 legend)
 - b. Secondly, while the data presented in Figure 1 highlight a potential role of IL22 in treatment response, and the following figures evidence a detrimental role of IL22 in IBD, there is still a disconnection between the two. As the IL22 ES is based on the response genes of IL22, the same effect could potentially be achieved by looking at other cytokines with similar response genes. This alternative hypothesis is supported by the low correlation between IL22 expression and IL22 ES (Supplementary Figure 5). It is entirely possible that the response to ustekinumab is reduced by the increased expression of any inflammatory process, a hypothesis supported by IL23A, IL-6, IL1beta, and TNFSF15 levels (Fig 2D). If the authors intend to present IL22 as the most likely candidate producing this non-response, they should at the very least present the same stratification in Figure 1 for the ES of the top response genes of the other cytokines presented in Figure 3.
 - c. If the importance of IL22 compared to other cytokines cannot be stated, Figure 1 would primarily serve to build the hypothesis that IL22 is detrimental in IBD, whereas the main findings of the paper would be that IL22 has a negative effect on IBD via neutrophils (and the text should be changed to reflect this). However, the novelty of these findings might be compromised.
2. The assumption that ILCs do not produce the measured chemokines in the co-cultures should be addressed in the article. The authors' response in the rebuttal is not convincing to rule out an expression of these chemokines in ILCs. The claim should be supported by data generated in the system used by the authors or should be addressed as an assumption rather than a fact. This reviewer is ok if the authors consider the possibility to tone down their conclusions.
3. The correlation between IL22 ES and UC1/UC2 ES from Czarnewski et al. is very interesting, especially as neutrophils have been highlighted in both studies. This reviewer believes that this confirming analysis and the novel mechanistic insights (through IL-22) provide an important novel aspect to the manuscript. It is surprising that the authors have not discussed this and have instead only focused on other therapies. The authors should include these analyses/data in the main figures and text.
4. The authors do not provide a clear rationale for looking at neutrophils specifically. Why were monocytes not considered? Although this reviewer understands that we can follow hints in a biased way, this should be clarified in the text, so the readers understand that other cell types might also play a role.

Minor issues

Results

Paragraph 1 (The IL22 responsive transcriptional network predicts...)

- GSE20971 is written as GEO20971. Also, it is in good form to at some point cite the original

papers for each GSE dataset used, when applicable. It is also good to add n for each study, either in legends or in methods.

- Phrasing suggesting “prediction” should be changed (see general comments): “Enrichment scores in colonic tissue could differentiate responders and non-responders”

Paragraph 2 (Patient stratification according to...)

- The text has not changed to reflect the changes in the figures (added significance and correlation matrix). It still says that there is “little difference” in IL23 expression although it is significant (it is indeed a “small” difference, but “little” difference suggests non-significance). Also, it still says that the increase is most notable in IL1B, which is not strongly supported by the figure. On the other hand, the correlation is the strongest for IL1B but this is not addressed. However, even there, the difference between IL1B and IL6 is small, so highlighting IL1B over IL6 is not really justified by the figures.

Paragraph 3 (IL22 regulates expression of...)

- Reference to Fig 3I is missing.

Paragraph 4 (IL22 mediated remodeling...)

- When discussing Fig S9C, it is stated that there is “time and dose dependent induction of CXC-family chemokine transcripts” when it only Cxcl5 has been tested.

Paragraph 6 (IL22 mediated induction...)

- The immunohistochemistry results only refer to the active UC samples, whereas all samples express all proteins. Please indicate this and highlight any differences by referring to the quantifications.

- The description of results from ref 28 is difficult to understand. At several points, the authors refer to “failed upregulation” whereas it is simply a lower expression compared to the control. Phrasing it as “failed upregulation” suggests that there is another sample group from which the control samples have upregulated expression whereas the mutant remained at the baseline.

- The following sentence is confusing: “In Stat3DIEL mice there was a failure of upregulation of several canonical IL22-regulated genes, such as Reg3b, Reg3g, Fut2 and Socs3, consistent with STAT3 being required for the induction of these transcripts in the epithelium.” “Consistent with” suggests that the first piece of information is supported by the second, but here it is the same information repeated (the data describes that Stat3 was necessary for these genes).

- There is no reference to Figure 6E.

Discussion

- Phrasing suggesting prediction should be changed (see general comments): “neutrophils as a functionally and prognostically important pathogenic pathway in UC”; “IL22 responsive transcriptional footprints [...] could differentiate patients”

Figures

Fig 1

Please add legend (red, blue, green) in figure as well.

Fig 2

The legend doesn’t reflect the changes made in the figure (A&B). The legend still does not explain the figure - it describes results, not methods (A,B,C). Please describe briefly how the figures were produced (e.g. 2b) “Genes belonging to “Migration of cells” pathway that are altered after IL22 treatment in colonic organoids, as identified using IPA downstream analysis. Genes have been further subdivided into functional categories. Symbols denote type of protein, color denotes regulation and color of line denotes relationship.”)

The changed scale and added significance in 2D are improvements to the figure, but the y-axis should still start at 0.

Fig 3

As mentioned in the first revision, panels C and I have low resolution. Please change the figures to better resolution ones.

Fig 4

Add n for each experiment.

Fig 5

Panel E says tests are performed against TRUC but it doesn't seem to be (and logically shouldn't be) the case for the blue bar (rescue).

Fig 6

It is not clear from the figure which species the results are from. Please indicate in the figure which experiment the data relate to.

A: The images do not look representative of the scores shown in Supplementary Figure 11. Please select images that are close to the group average.

E: The figure is not representative of the expected statistical test. The rationale for not showing the untreated group is unclear, and the result of only one test is shown when there would be three tests (between each pair of groups) performed. Also, were results normalized to untreated before or after testing? As normalization before the test would lead to an artificial standard deviation of 0 in the untreated group, this should be avoided. It appears that the rescued group results are all exactly 0, which is also surprising. Is $p=0.019$ true for both CXCL1 and CXCL5 (as indicated in legend)?

S1

The legend title overstates the result ("predicts", see general comments)

S9

There is no reference to C in the text. It is also not stated which dose was used, as requested by reviewer 3.

S10

Same as Fig 5 - Legend says that tests are performed against TRUC but it doesn't seem to be (and logically shouldn't be) the case for the blue bar (rescue).

S11&12

Graph axis says cells/nm². The two figures would benefit from being combined into one figure (both are already labeled as S11). The data is presented in different orders in figures compared to legend. It would be easier to read the graphs if the p-values were written in the graphs instead of long sentences in the legend

Reviewer #2 (Remarks to the Author):

Thanks for implementing many improvements. My comments have been sufficiently addressed. I feel that including a Figure related to organoid morphology will not be central to the manuscript's message and given the amount of suppl. Figures included.

Peter Hasselblatt

Reviewer #3 (Remarks to the Author):

Thank you.

The authors have addressed all my concerns.

REVIEWER COMMENTS

Reviewer #1 (Remarks to the Author):

In this revised version, Pavlidis and colleagues followed the reviewer's suggestions and removed entirely the IL23 data, and add new data that reinforces the findings described in the first version, notably concerning the role of IL22 in neutrophil infiltration.

The revised version provides strong evidence that IL22, which is counterintuitively associated with detrimental effects in IBD patients, promotes the induction of chemokines associated with neutrophil migration. However, the data claiming that IL-22 transcript levels predict responsiveness to biological is rather less convincing. Thus, this reviewer considers that claims made by the authors are not supported by the data.

Thank you for all your comments and your detailed review. We have now comprehensively addressed the points raised, and we believe that the manuscript has now been substantially improved. In keeping with the opinion of the reviewers, the relationship between IL22 responsive transcripts and prediction of response to ustekinumab has now been softened, and defined as an association (see below).

Major issues

1. There is still an issue in how the importance of IL22 in patients is being presented.

a. Firstly, several instances in the text (including the title) suggest that the IL22 ES has predictive power, but this is not supported by the data. Since the authors noted in the rebuttal that the score is not being proposed as a biomarker, it would suffice to change the title and text to reflect that the relation is associative, not predictive. (Title, abstract, pages 7 and 17, and S1 legend)

We agree. Since we are proposing that IL22 responsive genes are associated with poor response to ustekinumab (rather than being used as a biomarker to predict response), and accordingly as suggested, we have changed the language used.

b. Secondly, while the data presented in Figure 1 highlight a potential role of IL22 in treatment response, and the following figures evidence a detrimental role of IL22 in IBD, there is still a disconnection between the two. As the IL22 ES is based on the response genes of IL22, the same effect could potentially be achieved by looking at other cytokines with similar response genes. This alternative hypothesis is supported by the low correlation between IL22 expression and IL22 ES (Supplementary Figure 5). It is entirely possible that the response to ustekinumab is reduced by the increased expression of any inflammatory process, a hypothesis supported by IL23A, IL-6, IL1beta, and TNFSF15 levels (Fig 2D). If the authors intend to present IL22 as the most likely candidate producing this non-response, they should at the very least present the same stratification in Figure 1 for the ES of the top response genes of the other cytokines presented in Figure 3.

c. If the importance of IL22 compared to other cytokines cannot be stated, Figure 1 would primarily serve to build the hypothesis that IL22 is detrimental in IBD, whereas the main findings of the paper would be that IL22 has a negative effect on IBD via neutrophils (and the text should be changed to reflect this). However, the novelty of these findings might be compromised.

Thank you for your comment. Our intention is not to present IL22 as the sole candidate cytokine driving non-response in UC. Considering the overlap on transcriptional programmes regulated by canonical cytokines in human colonoids presented in figure 3 it is unlikely that a single cytokine is driving non-response. The clinical experience also corroborates this observation as therapeutics

move away from single cytokine targeting (i.e. antiTNF) to small molecules with broader downstream effects (i.e. JAK inhibitors).

Following your suggestion, we have now generated additional, new data looking to see whether the other cytokine enrichment scores associate with response to ustekinumab. There was no relationship at all between enrichment scores for IL17A and IL13 and response to ustekinumab. Interestingly, enrichment scores for IFN γ , the cytokine which most strongly overlapped with IL22, was also associated with response to ustekinumab, albeit slightly less than IL22 enrichment scores. TNF enrichment scores were only weakly associated with some, but not all outcomes. We have included these new data as important new supplementary figures. We had altered some aspects of the discussion in light of these data. We believe that although IL22 clearly regulates transcripts, including neutrophil-active chemokines, that are linked to poor outcomes, we also agree with the reviewers that our data are also consistent with more “complex” inflammation (or greater molecular severity of disease) being linked to poor outcomes. We have expanded the discussion to reflect this.

2. The assumption that ILCs do not produce the measured chemokines in the co-cultures should be addressed in the article. The authors’ response in the rebuttal is not convincing to rule out an expression of these chemokines in ILCs. The claim should be supported by data generated in the system used by the authors or should be addressed as an assumption rather than a fact. This reviewer is ok if the authors consider the possibility to tone down their conclusions.

As requested, we have toned down our conclusions to reflect that we are making an assumption that ILCs are not an important source of chemokines in this experimental system. Specifically, we have added the following sentence:

“This is based on the assumption that organoids were the chief source of chemokines in this experimental system. Although we did not formally test whether ILCs were a potential source of chemokines, previously published work has failed to detect the expression of these chemokines in colonic ILC3s²⁹.”

3. The correlation between IL22 ES and UC1/UC2 ES from Czarnewski et al. is very interesting, especially as neutrophils have been highlighted in both studies. This reviewer believes that this confirming analysis and the novel mechanistic insights (through IL-22) provide an important novel aspect to the manuscript. It is surprising that the authors have not discussed this and have instead only focused on other therapies. The authors should include these analyses/data in the main figures and text.

Thank you for your point, we agree that this is a relevant and important finding. We have added the correlation between the UC1UC2 gene set and the IL22 ES in main figures with a new paragraph detailing the results. Likewise, we have now included an additional section in the discussion to further explore and contextualise this association.

4. The authors do not provide a clear rationale for looking at neutrophils specifically. Why were monocytes not considered? Although this reviewer understands that we can follow hints in a biased way, this should be clarified in the text, so the readers understand that other cell types might also play a role.

The motivation for looking at neutrophils and neutrophil-active chemokines was data driven (in unsupervised analyses). Biological pathway analysis (canonical pathway analysis) identified “Granulocyte Adhesion and Diapedesis” as the most significantly enriched pathway in patients with

high IL22 enrichment scores ($P < 1.8 \times 10^{-29}$, Supplementary Figure 3). Likewise, Downstream effects/causal effects analysis identified cell migration among the most prominently activated. We had also generated a protein-protein interaction network based on DEGs regulated by IL22, which again highlighted neutrophil acting chemokines. This was previously a supplemental figure. To ensure the reader is clearer about the narrative, we have now brought this figure into the main figure panel of Figure 3. We have additionally adjusted the text to clarify this notion to the reader.

Minor issues

Results

Paragraph 1 (The IL22 responsive transcriptional network predicts...)

Changed to 'is associated with'

- GSE20971 is written as GEO20971. Also, it is in good form to at some point cite the original papers for each GSE dataset used, when applicable. It is also good to add n for each study, either in legends or in methods.

GEO changed to GSE, all studies are referenced, and n added in methods.

- Phrasing suggesting "prediction" should be changed (see general comments): "Enrichment scores in colonic tissue could differentiate responders and non-responders"

Paragraph 2 (Patient stratification according to...)

We have changed references to "prediction", as we agree that this work is not primarily focussed on biomarker discovery. However, we respectfully disagree that removing all references to the link between IL22 responsive transcripts, neutrophils and clinical outcomes to treatment with ustekinumab need to be removed (beyond just mentioning association). In patients with high IL22 ES, there is an important reduction in the rates of clinical remission (0.29 fold), mucosal healing (0.38 fold) and deep remission (0.21 fold) in comparison with patients with low IL22 ES. Although we are not making a major play about this being a predictive biomarker, it would also be incorrect to say that it doesn't differentiate response to ustekinumab. Notably, serum IL22 concentration - which is being developing as a companion diagnostic to stratify anti-IL23 therapy (Sands Gastro 2018), can differentiate responders and non-responders to anti-IL23 therapy in a comparable way (0.44 reduction in clinical response in CD patients). We believe that 'differentiate' and 'stratify' is appropriate language in this context (and these are also terms regularly used in similar studies, including the Czarnewski paper cited by the reviewer).

- The text has not changed to reflect the changes in the figures (added significance and correlation matrix). It still says that there is "little difference" in IL23 expression although it is significant (it is indeed a "small" difference, but "little" difference suggests non-significance). Also, it still says that the increase is most notable in IL1B, which is not strongly supported by the figure. On the other hand, the correlation is the strongest for IL1B but this is not addressed. However, even there, the difference between IL1B and IL6 is small, so highlighting IL1B over IL6 is not really justified by the figures.

Changed 'little' for 'small'. Amended text, reporting now both IL1B and IL6 as upstream regulators with higher expression.

Paragraph 3 (IL22 regulates expression of...)

- Reference to Fig 3I is missing.

Added now in relevant part of the text.

Paragraph 4 (IL22 mediated remodeling...)

- When discussing Fig S9C, it is stated that there is “time and dose dependent induction of CXC-family chemokine transcripts” when it only Cxcl5 has been tested.

Changed to *Cxcl5*

Paragraph 6 (IL22 mediated induction...)

- The immunohistochemistry results only refer to the active UC samples, whereas all samples express all proteins. Please indicate this and highlight any differences by referring to the quantifications.

Amended as per instruction

- The description of results from ref 28 is difficult to understand. At several points, the authors refer to “failed upregulation” whereas it is simply a lower expression compared to the control. Phrasing it as “failed upregulation” suggests that there is another sample group from which the control samples have upregulated expression whereas the mutant remained at the baseline.

Changed to ‘lower expression’

- The following sentence is confusing: “In Stat3DIEL mice there was a failure of upregulation of several canonical IL22-regulated genes, such as Reg3b, Reg3g, Fut2 and Socs3, consistent with STAT3 being required for the induction of these transcripts in the epithelium.” “Consistent with” suggests that the first piece of information is supported by the second, but here it is the same information repeated (the data describes that Stat3 was necessary for these genes).

Changed to ‘no expression’

- There is no reference to Figure 6E.

Now added

Discussion

- Phrasing suggesting prediction should be changed (see general comments): “neutrophils as a functionally and prognostically important pathogenic pathway in UC”; “IL22 responsive transcriptional footprints [...] could differentiate patients”

We have addressed this comment in our response to a previous point.

Figures

Fig 1

Please add legend (red, blue, green) in figure as well.

Now added

Fig 2

The legend doesn’t reflect the changes made in the figure (A&B). The legend still does not explain the figure - it describes results, not methods (A,B,C). Please describe briefly how the figures were produced (e.g. 2b) “Genes belonging to “Migration of cells” pathway that are altered after IL22 treatment in colonic organoids, as identified using IPA downstream analysis. Genes have been further subdivided into functional categories. Symbols denote type of protein, color denotes regulation and color of line denotes relationship.”)

The changed scale and added significance in 2D are improvements to the figure, but the y-axis should still start at 0.

Have amended legend as per your helpful instruction, now describing methods. Figure D y axis updated to start at 0.

Fig 3

As mentioned in the first revision, panels C and I have low resolution. Please change the figures to better resolution ones.

It appears that resolution dropped with file conversion when collating the final file, have now addressed, thank you.

Fig 4

Add n for each experiment.

Now added

Fig 5

Panel E says tests are performed against TRUC but it doesn't seem to be (and logically shouldn't be) the case for the blue bar (rescue).

Legend has been amended with appropriate statistical test and explanation of comparisons.

Fig 6

It is not clear from the figure which species the results are from. Please indicate in the figure which experiment the data relate to.

A: The images do not look representative of the scores shown in Supplementary Figure 11. Please select images that are close to the group average.

We have now amended with new images as per your instruction.

E: The figure is not representative of the expected statistical test. The rationale for not showing the untreated group is unclear, and the result of only one test is shown when there would be three tests (between each pair of groups) performed. Also, were results normalized to untreated before or after testing? As normalization before the test would lead to an artificial standard deviation of 0 in the untreated group, this should be avoided. It appears that the rescued group results are all exactly 0, which is also surprising. Is $p=0.019$ true for both CXCL1 and CXCL5 (as indicated in legend)?

Have now redrawn the figure. Values were normalised to the untreated group's mean and ANOVA with Tukey's multiple test correction was performed.

S1

The legend title overstates the result ("predicts", see general comments)

Have now amended, 'predict' has been replaced by 'associates.'

S9

There is no reference to C in the text. It is also not stated which dose was used, as requested by reviewer 3.

Legend has been amended and dose added.

S10

Same as Fig 5 - Legend says that tests are performed against TRUC but it doesn't seem to be (and logically shouldn't be) the case for the blue bar (rescue).

Legend has been amended

S11&12

Graph axis says cells/nm². The two figures would benefit from being combined into one figure (both

are already labeled as S11). The data is presented in different orders in figures compared to legend. It would be easier to read the graphs if the p-values were written in the graphs instead of long sentences in the legend

Amended as per suggestion

Reviewer #2 (Remarks to the Author):

Thanks for implementing many improvements. My comments have been sufficiently addressed. I feel that including a Figure related to organoid morphology will not be central to the manuscript's message and given the amount of suppl. Figures included.

Peter Hasselblatt

Thank you Prof Hasselblatt for taking the time to review our manuscript and all your insightful comments.

Reviewer #3 (Remarks to the Author):

Thank you.

The authors have addressed all my concerns.

Thank you for taking the time to review our manuscript and all your insightful comments

REVIEWERS' COMMENTS

Reviewer #1 (Remarks to the Author):

The authors have now addressed all our concerns.